# Can Pre-Trained Text-to-Image Models Generate Visual Goals for Reinforcement Learning?

**Jialu Gao**[1*],  **Kaizhe Hu**[1,2,3*],  **Guowei Xu**[1],  **Huazhe Xu**[1,2,3]

[1] Tsinghua University   [2] Shanghai Qi Zhi Institute   [3] Shanghai AI Lab

`gaojialululu@gmail.com, huazhe_xu@mail.tsinghua.edu.cn`

## Abstract

Pre-trained text-to-image generative models can produce diverse, semantically rich, and realistic images from natural language descriptions. Compared with language, images usually convey information with more details and less ambiguity. In this study, we propose Learning from the Void (LfVoid), a method that leverages the power of pre-trained text-to-image models and advanced image editing techniques to guide robot learning. Given natural language instructions, LfVoid can edit the original observations to obtain goal images, such as "wiping" a stain off a table. Subsequently, LfVoid trains an ensembled goal discriminator on the generated image to provide reward signals for a reinforcement learning agent, guiding it to achieve the goal. The ability of LfVoid to learn with zero in-domain training on expert demonstrations or true goal observations (the void) is attributed to the utilization of knowledge from web-scale generative models. We evaluate LfVoid across three simulated tasks and validate its feasibility in the corresponding real-world scenarios. In addition, we offer insights into the key considerations for the effective integration of visual generative models into robot learning workflows. We posit that our work represents an initial step towards the broader application of pre-trained visual generative models in the robotics field. Our project page: `LfVoid.github.io`.

## 1 Introduction

What is the simplest way to provide guidance or goals to a robot? This question is answered multiple times: a set of expert demonstrations [1, 2, 3, 4], goal images [5, 6, 7], or natural language instructions [8, 9]. However, these answers either require a laborious and sometimes prohibitive effort to collect data, or contain ambiguity across modalities. The desire to alleviate the challenges begs the question: can we generate goal images from natural language instructions directly, without physically achieving the goals for robots?

Large text-to-image generative models [10, 11, 12, 13] have achieved exciting breakthroughs, demonstrating an unprecedented ability to generate plausible images that are semantically aligned with given text prompts. Trained on extensive datasets, these models are thought to possess a basic understanding of the world [14]. Therefore, we aim to harness the power of these large generative models to provide unambiguous visual goals for robotic tasks without any in-domain training.

In the past, a common approach to leverage off-the-shelf large generative models in robot learning involves using these models for data augmentation on expert datasets to improve policy general-

---

*equal contribution

ization [15, 16, 17]. Despite the success, these methods still rely on human demonstrations, not generative models, as the main source of guidance.

DALL-E-Bot [18], an early attempt to utilize web-scale generative models in a zero-shot manner for guiding manipulation tasks, utilizes DALL-E 2 [10] to generate goal images for object rearrangement tasks. However, the generated images are often too diverse and largely disagree with real-world scenarios, requiring segmentation masks for object matching and a rule-based transformation planner to close the visual discrepancy.

The recent uprise of language-based image editing methods [19, 20, 21] provides a new paradigm: editing the generated images to align with language descriptions, such as "wiping a stain off a table", while keeping the visual appearance of the other objects and the background roughly unchanged. However, the edited images are usually examined by visual appearance rather than embodied tasks.

In this work, we introduce Learning from the Void (LfVoid), a method that uses image editing techniques on pre-trained diffusion models to generate visual goals for Reinforcement Learning (RL). LfVoid contains a unique image editing pipeline capable of performing appearance-based and structure-based editing to generate goal images according to language instructions. With these goal images, LfVoid improves upon example-based RL methods [5, 6] to solve several robot control tasks without the need for any reward function or demonstrations.

We evaluate LfVoid on three simulated environment tasks and the corresponding real-world scenarios. Empirical results show that LfVoid can generate goal images with higher fidelity to both the editing instructions and the source images when compared to existing image editing techniques, leading to better downstream control performance in comparison to language-guided and other image-editing-based methods. We also validate the feasibility of LfVoid in real-world settings: LfVoid can retrieve meaningful reward signals from generated images comparable to those from true goal images. Based on these observations, we discuss key considerations for the current generative model research when aiming for real-world robotic applications.

Our contributions are threefold: First, we propose an effective approach to leverage the knowledge encapsulated in large pre-trained generative models and apply it in a zero-shot manner to guide robot learning tasks. Second, we provide empirical evidence that suggests LfVoid-edited images provide guidance more effectively than other methods including the direct usage of text prompts. Lastly, we identify the existing gap between the capabilities of current text-conditioned image generation models and the demands in robotic applications, thus shedding light on a potential direction for future research in this domain.

## 2   Related work

**Image editing with diffusion models.**   Recent work in text-to-image diffusion models has demonstrated a strong ability to generate images that are semantically aligned with given text prompts [22, 23, 10, 11, 12, 24, 25]. Based on these generative models, Prompt-to-Prompt proposes to perform text-conditioned editing through the injection of the cross-attention maps. Imagic [20] proposes to perform editing by using interpolation between source and target embeddings to generate images with edited effects. InstructPix2Pix [26] demonstrates another approach for training a diffusion model on the paired source and target images and the corresponding editing instructions. Apart from editing techniques, several methods focusing on controlled editing have been proposed. Textual-Inversion [27] and DreamBooth [28] both aim to learn a special token corresponding to a user-specified object and at inference time can generate images containing that object with a diverse background. Directed Diffusion [21] focuses on object placement generation, which can control the location of a specified object to reside in a given area.

**Visual RL with implicit rewards.**   Standard reinforcement learning methods require an explicit reward function to guide the agent towards desired goal states, yet these reward functions may not always be available or may not capture the essence of the task at hand. One possible way of learning without access to explicit reward functions is using self-supervised learning to learn a representation where the distances can be used as reward signals [29, 30]. Another line of work uses example-based RL methods that aim to utilize the goal states to guide the learning process. VICE, proposed by Fu et al. [5], establishes a general framework for learning from goal observations: it trains a discriminator with the observations in the replay buffer as negative samples and goal images

as positive samples. The positive logits for new observations are used as a reward to guide the agent toward the goal observation. Building upon this, Singh et al. [6] introduce VICE-RAQ, integrating the VICE algorithm with active querying methods through periodically asking a human to provide labels for ambiguous observations. In addition, it proposes a label-smoothing technique for a better reward shaping. Other works [31, 32] also explore ways to improve the reward shaping of VICE as well. In a parallel direction, Eysenbach et al. [7] develop Recursive Classification of Examples (RCE), a method that learns the Q function of actor-critic methods directly from the goal observations.

**Large generative models for robot control.** Current application of large generative models in robotic tasks mainly focuses on using Large Language Models for planning [8, 9, 33, 34, 35] or learning a language conditioned policy [36, 37], while the application of pre-trained image generative models for robotics tasks are limited. A common way of leveraging large-scale diffusion models for robot learning is by using diffusion models to perform data augmentation on training data, as suggested in CACTI [15], GenAug [16] and ROSIE [17]. Another line of work suggests using diffusion models to generate plans to solve robotics tasks [38, 39, 40]. More recently, video generation and editing techniques have also been used with imitation learning to guide robot policies [41, 42]. More specifically, UniPi[42] trains a text-to-video generation model on web-scale datasets and expert demonstrations to generate image sequences for planning and inverse modeling. The most related work is DALL-E-Bot [18], which uses the DALL-E 2[10] model to generate goal plans and performs object rearrangement according to these goals. However, unlike our work, DALL-E-Bot does not apply image editing techniques for goal generation and requires rule-based matching and predefined pick-and-place actions to bridge the visual gap between generated images and true observations.

# 3 Method

In this study, we aim to provide zero-shot visual goals to reinforcement learning agents for manipulation tasks using only text prompts. Our approach utilizes the information grounded in large-scale pre-trained visual generative models to create semantically meaningful visual goals. We first generate a synthetic goal image dataset from raw observations using image editing techniques, then employ example-based visual reinforcement learning that is optimized for our task with the generated dataset.

The structure of this section unfolds as follows: Section 3.1 elucidates the image editing techniques and adjustments applied to create goal images from text prompts and initial image observations, as depicted in Figure 1(a). Section 3.2 discusses the execution of example-based visual reinforcement learning using the generated goal images, as outlined in Figure 1(b).

## 3.1 Visual goal generation

In the first phase of our methodology, we edit and generate goal images from raw observations based on natural language guidance provided by humans. Given a source prompt $\mathcal{P}$, a source image $x_{src}$, and an editing instruction, LfVoid synthesizes a target image $x_{tgt}$ using a pre-trained Latent Diffusion Model (LDM) [12]. We consider two types of editing instructions: either a target prompt $\mathcal{P}^*$ describing the appearance changes in the target image, or a bounding-box region $\mathcal{B}$ and a set of tokens $\mathcal{I}$ corresponding to the object to be relocated when structural changes are needed.

To provide sufficient visual guidance for downstream reinforcement learning tasks, the generated goal images should highlight the visual changes while preserving the irrelevant scene as much as possible, which is a challenging requirement even for state-of-the-art image editing techniques. To this end, we integrate a number of different techniques in the visual goal generation pipeline of LfVoid, consisting of a feature extracting module, an inversion module, and an editing module, as outlined in Figure 1(a). Following conventional notations, we denote $x_t$ as the synthesized LDM image at time step $t \in \{T, T-1, ..., 0\}$, where $T$ is the total diffusion time steps. Specifically, $x_T$ denotes the Gaussian noise, and $x_0$ denotes the generated image. The techniques used in each module and their purpose are described as follows.

### 3.1.1 Feature extracting module

To ensure high fidelity to the source image $x_{src}$, we learn a unique token $sks$ that encapsulates the visual features of objects within $x_{src}$ as in DreamBooth [28]. This process involves optimizing the diffusion model parameters along with the special token $sks$, using a set of images that contain the

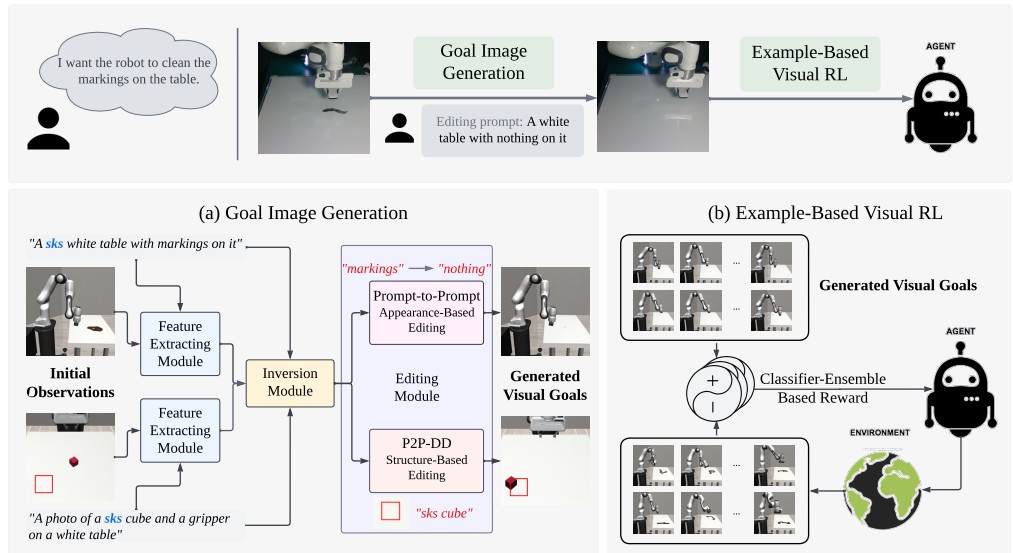

Figure 1: **An overview of LfVoid.** The goal of LfVoid is to learn robot policies requiring only language descriptions from humans. LfVoid consists of two parts: (a) Goal image generation, where we apply image editing on the initial observations according to different editing instructions to obtain a visual goal dataset; (b) Example-Based Visual RL, where we perform reinforcement learning on the generated dataset to achieve the desired goal image in various environments.

target object. As a result, we are able to derive a specialized model that can accurately retain key details of $x_{src}$, such as the color and texture of a cube to be positioned, or the shape of a Franka robot arm for manipulation. As later demonstrated in our experiments, this module substantially boosts the resemblance in details of our edited images to the corresponding source images.

### 3.1.2 Inversion module

After employing the special token $sks$ to capture the essential features of the source scene, we invert the provided source image $x_{src}$ to a diffusion process. A simple inversion technique using the Denoising Diffusion Implicit Model (DDIM) [23] sampling scheme calculates the samples $x_T, x_{T-1}, ..., x_0$ by reversing the ODE process. However, the inverted image $x_0$ obtained this way often deviates from $x_{src}$ due to cumulative errors. Null-text inversion [43] aims to mitigate this discrepancy by fine-tuning the "null-text" embedding used in the classifier-free guidance [44] $\Phi_t$ for each $t \in \{T, T-1, ..., 1\}$ to control the generation process. Thus, with the initial noise sample $x_T$ obtained through DDIM inversion and the optimized "null-text" embedding $\Phi_T, \Phi_{T-1}, ..., \Phi_1$, the diffusion process is able to generate the image $x'_{src}$, an approximation of the source image $x_{src}$. Based on the diffusion process obtained with the inversion module, LfVoid can perform precise and detailed editing control on the source image, which will be discussed in Section 3.1.3.

### 3.1.3 Editing module

Depending on the specific requirements, we divide the editing tasks into two distinct categories: appearance-based and structure-based image editing. Appearance-based editing involves maintaining the structural layout of the image while altering certain visual aspects, such as cleaning the surface of a table or lighting up an LED bulb. In contrast, structure-based editing involves changes in the layout of the image, such as relocating objects within the image from one area to another.

**Appearance-based editing.** In tasks involving appearance-based editing, we employ the Prompt-to-Prompt editing control technique. When the Latent Diffusion Model (LDM) generates an image $x_0$ conditioned on a text prompt $\mathcal{P}$, the information encapsulated in $\mathcal{P}$ influences the diffusion process via the cross-attention layers [12, 11, 19]. Prompt-to-Prompt reveals that the spatial configuration

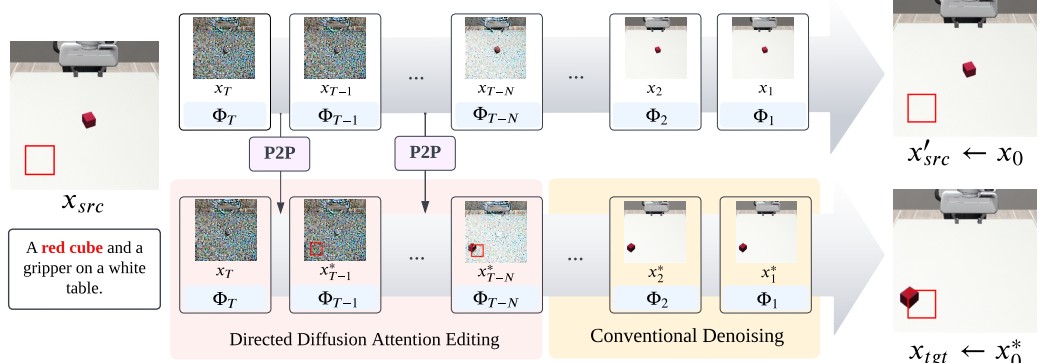

Figure 2: **Image editing with structural changes.** The details of P2P-DD technique used in the editing module of LfVoid to perform structural changes. LfVoid performs a combination of P2P attention map injection with Directed Diffusion attention editing on the diffusion process generating the target image $x_{tgt}$, based on the Gaussian noise $x_T$ and the optimized null-text embeddings $\Phi_T, \Phi_{T-1}, ..., \Phi_1$ obtained through the Inversion Module of LfVoid.

of the image $x_0$ largely depends on the cross-attention maps $M_t$ within these cross-attention layers, especially during the initial diffusion steps. Consequently, it suggests replacing the attention maps $M_t^*$ of the target image diffusion process with the attention maps $M_t$ from the source image diffusion process for the first $N$ time steps, beginning at time step $t = T$:

$$\text{P2PEdit}(M, M^*, t) = \begin{cases} M_t & \text{if } t > T - N \\ M_t^* & \text{otherwise} \end{cases} \tag{1}$$

This method ensures that the structure of the source image encapsulated in the attention maps $M_t$ is conserved in the target image in a more controlled manner, which is crucial in downstream RL tasks.

**Structure-based editing.** A notable limitation of Prompt-to-Prompt editing is the incapacity to spatially move existing objects across the image, i.e., implement structural changes. Therefore, we introduce a novel approach termed P2P-DD for structure-based editing tasks. P2P-DD combines Prompt-to-Prompt control with the idea of Directed Diffusion [21], extending its capability to perform object replacement.

Directed Diffusion [21] illustrates the possibility of achieving object replacement through direct editing of attention maps during the first $N$ time steps of the generation process. The method performs attention strengthening on the cross-attention maps corresponding to the tokens in $\mathcal{I}$ (recall that $\mathcal{I}$ is the token sets representing the object of interest), through calculating a Gaussian strengthening mask (SM) of the bounding-box region $\mathcal{B}$. It also performs attention annealing to the remaining area $\bar{\mathcal{B}}$ by applying a constant weakening mask (WM), and the two masks are weighted by a scalar $c$:

$$\text{DDEdit}(M_t, \mathcal{B}, \mathcal{I}) = \begin{cases} M_t \odot \text{WM}(\bar{\mathcal{B}}, \mathcal{I}) + c \cdot \text{SM}(\mathcal{B}, \mathcal{I}) & \text{if } t > T - N \\ M_t & \text{otherwise} \end{cases} \tag{2}$$

Our proposed P2P-DD method aims to encourage the background and other details of the image generated by Directed Diffusion with a higher resemblance to the source image. P2P-DD first injects cross-attention maps $M_t$ into $M_t^*$ as suggested by Prompt-to-Prompt. This will preserve the structure and background information from the source image, providing a decent starting point for attention map editing. Next, P2P-DD performs pixel value strengthening and weakening on the attention maps $M_t$ according to the bounding-box $\mathcal{B}$ and the tokens in $\mathcal{I}$, as suggested in the Directed Diffusion, to achieve the desired object placement:

$$\text{P2P-DDEdit}(M_t, M_t^*, \mathcal{B}, \mathcal{I}) = \text{DDEdit}(\text{P2PEdit}(M, M^*, t), \mathcal{B}, \mathcal{I}) \tag{3}$$

This combined editing control is only applied in the first $N$ diffusion time steps, i.e. $\{T, ..., T-N+1\}$. After the first $N$ time steps, we cease any control on attention maps and allow the target diffusion

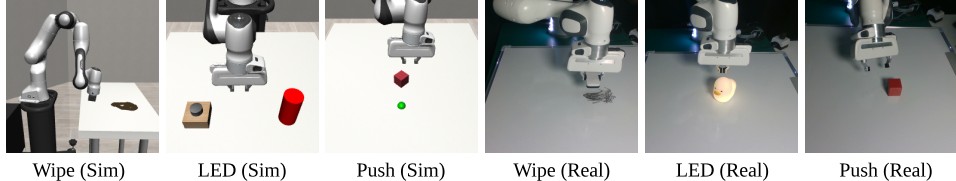

| Wipe (Sim) | LED (Sim) | Push (Sim) | Wipe (Real) | LED (Real) | Push (Real) |

Figure 3: **Visualization of robot manipulation tasks.** We evaluate LfVoid on three simulation tasks: Wipe, Push, and LED, as well as three real-robot environments correspondingly.

process to complete in a conventional denoising manner. The details of P2P-DD are shown in Figure 2. Attention visualization of the generation process can be found in Appendix A.1, and we provide a detailed description of the full algorithm in Appendix A.2.

## 3.2 Example-based visual reinforcement learning

We modify and extend the approach of VICE [5] to devise our method for example-based visual reinforcement learning. At the core of our process, we employ the image editing methods described in Section 3.1 on observations gathered from randomly initialized environments, producing a dataset of 1024 target images for each task. During training, these target images serve as positive samples, while the images sampled from the agent's replay buffer after a number of random exploration steps are treated as negative samples. We train a discriminator using these instances with the binary cross-entropy loss and use the output of the discriminator for new observations for the positive class as the agent reward.

The discriminator shares the CNN encoder with the reinforcement learning agent and performs classification over the output latent representations. To improve the reward shaping, we utilize the label mixup method [6], which performs a random linear interpolation of the 0-1 labels and their corresponding hidden vectors to obtain continuous labels between 0 and 1. We discover that restricting the negative instances from the recent portion of the replay buffer (the last 5%) promotes the discriminator's ability to discern subtle differences between the target and current observations. Additionally, we find that our method enjoys an ensemble of discriminators for classification results (i.e., RL rewards), which gives the agent more representative rewards.

For the reinforcement learning backbone algorithm, we use DrQ-v2 [45] for visual RL training. Based on Twin Delayed DDPG (TD3) [46], DrQ-v2 enhances image representation by applying random crop augmentation to the input images. Collectively, these refinements constitute our approach to example-based visual reinforcement learning, making the "learning" from an initial observation and language prompts possible.

## 4 Experiments

In this section, we present a comprehensive evaluation of LfVoid across both simulated environments and real-world robotic tasks. An illustration of each task can be found in Figure 3.

The environments we use are: 1) LED, where the robot reaches for a switch or touches the LED light directly to turn the light from red to green; 2) Wipe, where the robot needs to wipe out stains from the table; 3) Push, where the robot needs to push a red cube to a goal position indicated by a green dot. The simulated tasks are developed based on the Robosuite benchmark, while we provide corresponding real-world tasks for each environment. A full description of the environments is provided in Appendix B.1.

### 4.1 Goal generation

In this section, we evaluate the ability of LfVoid and other baseline methods to generate goal images according to two types of editing instructions: appearance-based editing and structure-based editing.

| | Input | Imagic | InstructPix2Pix | DALL E-2 | DDIM P2P | DreamBooth P2P | Null-text P2P | **LfVoid (Ours)** |

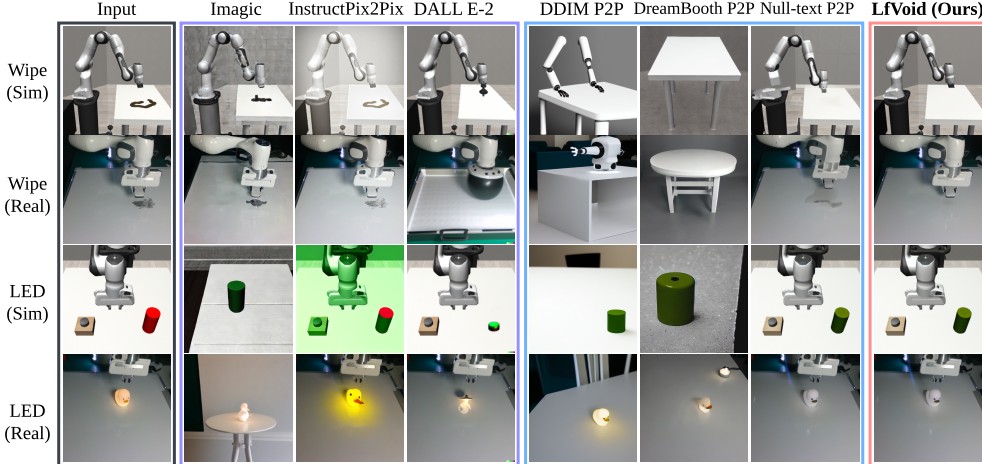

Figure 4: **Appearance-based goal image editing results.** We compare LfVoid against three recent image editing methods: Imagic, InstructPix2Pix, and DALL E-2, as well as three ablations of LfVoid. Note that InstructPix2Pix has different prompts because it requires an editing instruction rather than a description of the image. In both simulation and real-world settings, LfVoid can generate images that are better aligned with given text prompts while preserving the remaining source scene. See Appendix C for more examples.

**Baselines and ablations.** We select three recent image editing methods as baselines: Imagic [20], InstructPix2Pix [26], and mask-based DALL E-2 editing used in DALL-E-Bot [18](with masks provided by humans). For ablations, DDIM P2P removes both the feature extracting module and the inversion module, and DDIM DD additionally removes P2P in the editing module for structure-based editing. On top of DDIM DD, Null-text DD adds the inversion module back. For both appearance-based and structure-based editing tasks, DreamBooth P2P/P2P-DD ablates the inversion module, and Null-text P2P/P2P-DD ablates the feature extracting module.

**Appearance-based goal image editing results.** In Figure 4, we present the visual goal generation results of the Wipe and LED tasks, focusing on editing object appearances in both simulated and real-world environments. Imagic struggles to preserve the structure and details of the source image, and InstructPix2Pix frequently fails to perform local color editing, unable to locate the specific area indicated by the editing instruction. DALL E-2 is able to preserve the scenes outside the masks but is not capable of performing the desired edits within the masked area. For ablations, DDIM P2P and DreamBooth P2P-DD can mostly carry out the desired edits but often diverge largely from the source image. This demonstrates the importance of the inversion module in preserving the details and structures of the source image. Null-text P2P performs worse in the Wipe tasks compared to LfVoid, demonstrating the gains provided by the DreamBooth token. Overall, LfVoid demonstrates superior performance in terms of maintaining the appearance of unrelated parts while effectively editing specified regions: through directly applying control and editing on the attention maps, LfVoid is capable of locating the target area and performing the desired edit based on language instructions.

**Structure-based goal image editing results.** In Figure 5, we report the performance of LfVoid in editing tasks with structural changes (the Push task) in both simulated and real-world settings. For baselines, Imagic distorts the robot arm and table significantly and often introduces multiple objects into the image, rather than moving the existing one. InstructPix2Pix only changes the overall color of the image or replaces the gripper with a geometric body, failing to achieve the required object displacement. DALL E-2 also fails to relocate the object to the desired place. For ablative results, while DDIM DD and Null-text DD can successfully move objects to the lower-left corner, the background and the robot arm nearly disappear. Null-text P2P-DD improves the preservation of the source image's background, but the shape of the moved object is inconsistent with its original form. DreamBooth P2P-DD can preserve the shape of the moved object and relocation is successful, but

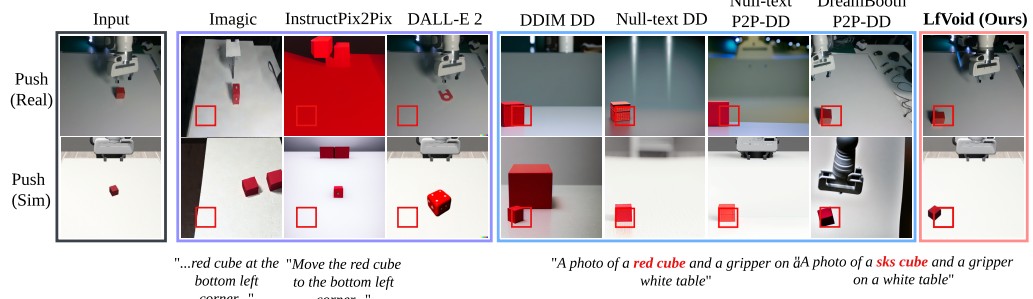

Figure 5: **Structure-based goal image editing results.** We compare LfVoid against Imagic, Instruct-Pix2Pix, and DALL E-2, as well as four ablations of LfVoid. Note that InstructPix2Pix has different prompts because it requires an editing instruction rather than a description of the image. LfVoid can successfully perform object displacement and preserve the background and details of the source image, while other methods fail to do so. See Appendix C for more examples.

the background is not consistent with the source image. Our method, when compared to all baselines, can perform the object displacement task successfully while better preserving the visual features of the object and the remaining scene. These experiments reveal the importance of each component in our method. The creative integration of different image editing techniques provides a powerful tool for image editing with structural changes, demonstrating a significant improvement over existing methods.

**Quantitative results.** In Table 1, we present the quantitative results of goal image generation. We manually sample 500 real goal images per task(only for evaluation purposes) and calculate the pair-wise LIPIS distance between the generated images and the real goal images. The lower distance represents higher image editing quality, and results show that LfVoid outperforms all the baselines and ablations. Additionally, we present an ablation study of the feature extracting module on the Wipe and Push task in Table 2, showing the performance gain brought by the DreamBooth token.

|  | Wipe(Sim) | LED(Sim) | Push(Sim) | Wipe(Real) | LED(Real) | Push(Real) |
|---|---|---|---|---|---|---|
| InstructPix2Pix | 0.23±0.11 | 0.41±0.06 | 0.34±0.17 | 0.18±0.04 | 0.45±0.06 | 0.53±0.08 |
| Imagic | 0.23±0.05 | 0.42±0.08 | 0.43±0.11 | 0.34±0.13 | 0.35±0.16 | 0.49±0.11 |
| DALL-E 2 | 0.10±0.04 | 0.25±0.07 | 0.20±0.06 | 0.39±0.08 | 0.20±0.07 | 0.38±0.03 |
| DreamBooth+Editing | 0.53±0.05 | 0.62±0.05 | 0.42±0.15 | 0.58±0.06 | 0.43±0.08 | 0.36±0.06 |
| DDIM+DD/P2P | 0.26±0.07 | 0.37±0.06 | 0.43±0.04 | 0.34±0.04 | 0.40±0.03 | 0.50±0.03 |
| Null-text+DD | N/A | N/A | 0.36±0.05 | N/A | N/A | 0.51±0.06 |
| LfVoid | **0.08±0.04** | **0.17±0.06** | **0.14±0.07** | **0.12±0.05** | **0.19±0.05** | **0.32±0.03** |

Table 1: **Quantitative results on goal image generation.** We calculate the pair-wise LIPIS distance between generated images and the manually created real goal images for evaluation purposes.

|  | Wipe(Sim) | Push(Sim) | Wipe(Real) | Push(Real) |
|---|---|---|---|---|
| LfVoid w/o DreamBooth | 0.10±0.04 | 0.24±0.06 | 0.18±0.06 | 0.39±0.06 |
| LfVoid | **0.08±0.04** | **0.14±0.07** | **0.12±0.05** | **0.32±0.03** |

Table 2: **Ablation study on DreamBooth.** We evaluate the improvement brought by the DreamBooth token through pair-wise LIPIS distance between generated images and manually created real goal images for evaluation purposes.

## 4.2 Example-based visual reinforcement learning

We now dive into the results obtained from applying example-based visual reinforcement learning using the goal images generated by LfVoid. We select three baselines for comparison: CLIP, IP2P,

and Real Goal. CLIP relies only on language guidance and does not require any synthesized goal images. It leverages the Contrastive Language-Image Pre-training score [47] of the task prompt and observations as a reward, and it aims to verify the effectiveness of translating prompts into images against raw language hints. IP2P uses InstructPix2Pix for generating the goal images and then runs the same example-based RL as in LfVoid. It allows us to compare the performance of LfVoid with a straightforward image editing technique. Real Goal is an upper-bound performance of our pipeline, where we manually create ideal goal images, such as erasing the stains directly in the simulator for the Wipe task. This strategy illustrates the potential of our algorithm, indicating the gap between the generated goals and the real ones, and demonstrates how the quality of generated images will influence the performance of downstream RL.

**Simulation results.** We report both episode reward (Figure 6) and numerical metrics (Table 3) on three simulation tasks. The training details can be found in Appendix B.3. We observe that the CLIP baseline achieves low rewards in all the tasks, showing that directly using language guidance through CLIP embeddings is not sufficient. Moreover, LfVoid consistently outperforms the IP2P baseline and gains comparable performance with the Real Goal upper bound, providing evidence that generated goal images with high fidelity to both the original scene and the editing instructions is crucial for successful deployment of downstream example-based RL.

**Real-world results.** In the more complex real-world settings, we aim to validate the feasibility of LfVoid through visualization of reward functions. As in our previous experiment, we generate goal images using InstructPix2Pix (IP2P) and LfVoid, and also manually collect a real goal dataset as an upper-bound (Real Goal). We then train discriminators on these datasets separately and use the outputs of the discriminators as rewards. In addition, we compare with language-based guidance (CLIP), which uses the distance of the CLIP embeddings between the task prompt and the observation images as rewards. We manually record five successful trajectories for each task and use these reward functions to obtain the reward curves. As illustrated in Figure 7, LfVoid can provide near monotonic dense reward signals comparable to those from Real Goal. On the contrary, the curves obtained by IP2P and CLIP fail to accurately track the progress of each task. This demonstrates that our method provides more instructive learning signals for real-world robotic tasks than either IP2P or CLIP.

|  | CLIP | InstructPix2Pix | Real Goal (Oracle) | Ours |
|---|---|---|---|---|
| Wipe (Cleaned Stains / Patch ) | 0.0±0.0 | 1.7±1.9 | 22.0±5.8 | **21.3±5.8** |
| LED (Success Rate / %) | 10.0±20.0 | 0.0±0.0 | 93.3±11.5 | **75.0±50.0** |
| Push (Success Rate / %) | 12.5±19.1 | 3.3±3.5 | 30.0±13.7 | **27.9±12.7** |

Table 3: **Numerical metrics of simulation tasks.** We report the success rate for LED and Push, and the number of stain patches cleaned for Wipe. Please refer to Appendix B.1 for details.

**Comparison to RL with imagined goals.** In a parallel direction, RL with imagined goals can also solve robot manipulation tasks without the need for human-specified reward functions and expert demonstrations [29, 30]. These methods first leverage self-supervised learning to learn a representation

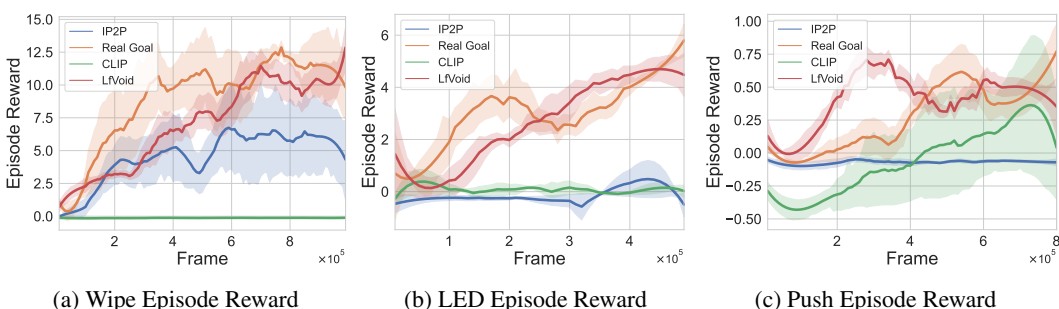

(a) Wipe Episode Reward     (b) LED Episode Reward     (c) Push Episode Reward

Figure 6: **Episode reward of simulation tasks.** The results of the CLIP baseline (CLIP), Instruct-Pix2Pix baseline (IP2P), using real goal image as upper bound (Real Goal), and LfVoid (Ours).

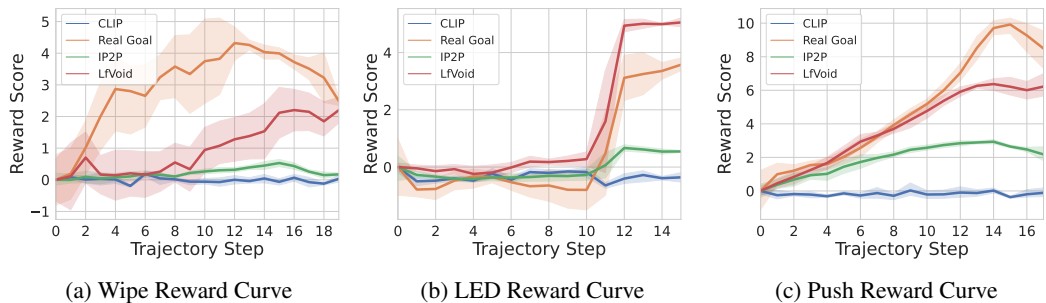

| (a) Wipe Reward Curve | (b) LED Reward Curve | (c) Push Reward Curve |

Figure 7: **Reward curve on successful trajectories for real-robot tasks.** Visualization of the reward function of LfVoid (Ours) compared with baselines. See Appendix B.4 for implementation details.

space of observations, and then sample imagined goals to train a goal-conditioned policy. However, unlike LfVoid, a real goal image is still required at test time to run the goal-conditioned policy. In Table 4, we compare the performance of LfVoid with two algorithms: Visual RL with Imagined Goals (VIG) [29] and Goal-Aware Prediction (GAP) [30], on the three simulation tasks. We observe that only LfVoid can solve the tasks with positive rewards, the other two methods fail in all tasks. Because both these methods use random policy for exploration, they may never reach the user-specified goal states during training. Therefore, they lack the knowledge required to fulfill the goals. This further demonstrates the significance of leveraging web-scale text-to-image models to provide informative visual guidance during training. Please refer to Appendix C.2 for more analysis.

|  | Push(Sim) | Wipe(Sim) | LED(Sim) |
| --- | --- | --- | --- |
| Visual rl with Imagined Goals (VIG) | -0.39±1.79 | -6.03±2.02 | -1.77±0.55 |
| Goal-Aware Prediction (GAP) | -0.07±0.91 | -5.84±1.00 | -0.39±0.11 |
| Learning from the Void (LfVoid) | **0.35±0.51** | **12.86±4.31** | **4.47±2.10** |

Table 4: **Comparison with RL using imagined goals.** We report the episode reward during test time and compare LfVoid with existing methods of visual RL using imagined goals.

## 5 Discussion

In this work, we present LfVoid, an effective approach for leveraging the knowledge of large-scale text-to-image models and enabling zero-shot reinforcement learning from text prompts and raw observation images. We have identified and tackled numerous challenges that arise when applying state-of-the-art image editing technologies to example-based RL methods. Our work highlights the potential for the adaptation and application of image-generation techniques in the realm of robotics. Our findings not only enhance the understanding of image editing for robotic applications but also provide a clear direction for the image generation community to address real-world challenges.

Although our proposed method has shown promising results, we acknowledge that it is not without limitations. While LfVoid has succeeded in generating goal images to train example-based RL policies, a certain level of prompt tuning is needed in order to achieve optimal editing performance (see Appendix B.2), and we refer the readers to Appendix D for an analysis of failure cases.

We would also like to highlight that there exists a gap between the ability of text-to-image generation models and the need for robot learning. For example, large diffusion models exhibit poor under-standings of the spatial relationships between objects [48]; therefore, both the generative models and the editing methods struggle to handle object displacement solely through language prompts. The Directed Diffusion technique, while being able to perform movements of objects, requires a user-defined bounding box to achieve precise control. Furthermore, large generative models sometimes struggle to generate images that are considered valid under physics laws. When asked to pick up a bottle with a robot arm, the model simply stretches the shape of the arm to reach the bottle rather than changing the joint position. Lastly, AI-generated images inevitably introduce alterations to the details of objects. The robustness of current visual reinforcement learning algorithms to such changes remains an open question.

## Acknowledgment

HX is supported by National Key R&D Program of China (2022ZD0161700).

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
