# A   Algorithm details

## A.1   Attention visualization

**Prompt-to-Prompt editing.**   In the Prompt-to-Prompt editing algorithm [19], attention editing (P2PEdit), as defined in Equation 1, is applied on the cross-attention maps during first $T - N$ steps, where $T = 50, N = 10$ for Wipe. For visualization purposes, we calculate the average cross-attention maps of size $16 \times 16$ at time step $t = 50, 40, 30, 20, 10, 0$ during the source image diffusion process as well as the target image diffusion process, and present the results in Figure 8. We can observe that in time steps $t = 50, 40, 30, 20, 10$, the attention maps in the upper row (source image diffusion) are very similar to that of the bottom row (target image diffusion): this is because the cross-attention maps from the upper row are injected to the bottom row, in order to preserve the structure of the figure. However, after the first 40 steps, we cease any control on the attention maps, therefore the attention maps of the bottom row significantly differ from the upper at time step $t = 0$. This is when the editing instruction of replacing "markings" with "nothing" is performed, as indicated by the red box in Figure 8.

**Directed Diffusion editing.**   In Figure 9, we present the visualization of Directed Diffusion [21] used in the Push task. Recall that in Directed Diffusion, we perform value strengthening in the bounding-box area and weakening the other area of the attention maps during the first $T - N$ diffusion steps. For the Push task, we set $T = 50, N = 30$. At step $t = 50, 40$, attention strengthening is performed at tokens corresponding to the word "red" and "cube", as well as the trailing tokens at the end of the sentence. After the first 20 steps, we let the diffusion process finish in a conventional denoising manner. As shown in the attention maps from time-step $t = 30, 20, 10, 0$, the strengthening effect is preserved during the later diffusion steps, leading to a successful displacement. However, because we anneal the remaining area outside the bounding box, the background information of the source image is lost during the attention editing (such as the appearance of the gripper), therefore the background of the generated target image differs largely from the source image.

**P2P-DD editing.**   We propose the P2P-DD algorithm, a combination of Prompt-to-Prompt and Directed Diffusion techniques to achieve structure-based editing for the Push task. As defined in Equation 3, P2P-DD runs two diffusion processes in parallel. As shown in Figure 10, the upper row represents the attention map of the source image diffusion process, and the bottom row represents that of the target image. Instead of directly performing strengthening and weakening on the attention maps, P2P-DD first injects the attention maps from the source diffusion process and then performs editing on these injected attention maps. In this way, the details of the source image can be better preserved: the gripper in the background is preserved in the attention maps of later time steps, as indicated by the blue box in Figure 10.

## A.2   Full algorithm

The complete visual goal generation algorithm is presented in Algorithm 1, and the complete example-based visual RL algorithm is provided in Algorithm 2.

Following conventional notations, we denote $x_t$ as the synthesized LDM image at time-step $t \in \{T, T - 1, ..., 0\}$, where $T$ is the total diffusion steps. Specifically, $x_T$ denotes the Gaussian noise, and $x_0$ denotes the generated image. For notations used in Algorithm 1, $\Phi_t$ denotes the "null-text" embedding at step $t$, and $M_t$ denotes the attention maps at step $t$.

For the example-based RL part, starting from a dataset of edited target observation $\mathcal{D}^{edit}\{s_i^{edit}\}_{i=1}^{N_g}$. We use the current policy $\pi$ to interact with the environment and collect a dataset $\mathcal{D}^{env}$, and train an ensemble of classifiers $\{\Phi^i\}_{i=1}^{N_c}$ on the linearly mixuped batch $\mathcal{B}$ consists of samples $\{s_i^{env}\}_{i=1}^{\mathcal{B}}$ from $\mathcal{D}^{env}$ as negative class, and samples from the edited images $\{s_i^{edit}\}_{i=1}^{\mathcal{B}}$ as positive class. During training, we use DrQv2 as the base RL algorithm and use the positive logits from the classifier as the reward. The update of classifiers and the RL agent take place in turn and we only use the recent $\lambda$-portion of the replay buffer (denoted as $\mathcal{D}_\lambda^{env}$) to train the classifier. The pseudo-code of example-based RL can be found at Algorithm 2. And an introduction to the mixup technique we use is provided as follows:

**Algorithm 1** Visual Goal Generation of LfVoid

---

1: **Input**: A source prompt embedding $P$, a source image $x_{src}$, a diffusion model $DM(.)$, a text encoder $\phi(.)$, a target prompt $P^*$(if appearance-based), a bounding-box region $\mathcal{B}$ and a token set $\mathcal{I}$(if structure-based)
2: **Output**: A target image $x_{tgt}$

---

3: Set guidance scale $\omega = 1$
4: Compute the DDIM inversion results $x'_T, ..., x'_0$ over the source image $x_{src}$
5: Set guidance scale $\omega = 7.5$
6: Initialize $\bar{x}_T \leftarrow x'_T$, null-text embedding $\Phi_T \leftarrow \phi("")$
7: **for** t=T, T-1, ... 1 **do**
8:    **for** j=0, ... N-1 **do**
9:       $\Phi_t \leftarrow \Phi_t - \eta\nabla_\Phi||x'_{t-1} - x_{t-1}(\bar{x}_t, \Phi_t, \phi(P))||^2_2$
10:    **end for**
11:    Set $\bar{x}_{t-1} \leftarrow x_{t-1}(\bar{x}_t, \Phi_t, \phi(P))$, $\Phi_{t-1} \leftarrow \Phi_t$
12: **end for**
13: $x_T \leftarrow x'_T$, $x^*_T \leftarrow x_T$
14: **for** $t = T, T-1, ..., 1$ **do**
15:    $x_{t-1}, M_t \leftarrow DM(x_t, P, t, \Phi_t)$
16:    $M^*_t \leftarrow DM(x^*_t, P, t, \Phi_t)$
17:    **if** appearance-based editing **then**
18:       $M^*_t \leftarrow \text{P2PEdit}(M_t, M^*_t, t)$
19:       $x^*_{t-1} \leftarrow DM(x^*_t, P^*, t, \Phi_t)\{M \leftarrow M^*_t\}$
20:    **else**
21:       $M^*_t \leftarrow \text{P2P-DDEdit}(M_t, M^*_t, \mathcal{B}, \mathcal{I})$
22:       $x^*_{t-1} \leftarrow DM(x^*_t, P, t, \Phi_t)\{M \leftarrow M^*_t\}$
23:    **end if**
24: **end for**
25: $x_{tgt} \leftarrow x^*_0$
26: Return $x_{tgt}$

---

Consider $s_i$ and $s_j$ as any two inputs to the classifier, originating either from the replay buffer or the set of goal examples, with their associated labels $y_i$ and $y_j$. Mixup regularization employs linear interpolation to these input/output pairs to construct a synthetic training distribution as follows:

$$\tilde{s} = \lambda s_i + (1-\lambda)s_j, \quad \tilde{y} = \lambda y_i + (1-\lambda)y_j \tag{4}$$

Here, $\lambda \sim \text{Beta}(\alpha, \alpha)$ follows the Beta distribution. The mixup parameter $\alpha$ determines the degree of mixup, with higher $\alpha$ values implying a greater mixup level (i.e., the sampled $\lambda$ values are closer to 0.5 than 0). As illustrated by Zhang et.al. [49], mixup contributes to smoother transitions between distinct classes by promoting linear behavior.

## B   Implementation details

### B.1   Environment settings

**Wipe environment.** For the wipe simulation environment (Wipe Sim), we adopt the same setting from the official Robosuite Benchmark [51]. In this task, the agent is expected to clean the markings made up of 100 dark particles on a white table. A dense reward is given based on the distance of the wiper to the stains, the contact between the wiper and table, the particles being wiped, and the completion of the task. A penalty will also be given for collision or joint limit reached, large force, and large acceleration of the wiper. Since this dense reward cannot accurately represent how far along the task is completed, we also record the total number of particles being wiped in an episode and use that value for numerical results in Table 3.

**Push environment.** Based on the implementation of the Push task on other environments [52], we set up a "Push" environment in Robosuite. The target of this task is to use the robot arm and end-effector to push a red cube to a specific position. In the visual observations of this environment,

**Algorithm 2** Example-based Visual RL of LfVoid

---

1: **Input**: An edited dataset $\mathcal{D}^{edit}$, ensemble of classifiers $\{\Psi^i\}_{i=1}^{N_c}$, an DrQv2 agent $\pi$ and critic $Q_1, Q_2$, classifier training interval $t_{cls\_int}$, classifier training steps $t_{cls\_steps}$, label mixup hyper-parameter $\alpha$, training environment $e$, batch size $\mathcal{B}$

2: **Output**: Trained DrQv2 agent $\pi$

---

3: $s \leftarrow e.\text{reset}()$
4: **for** $t = T, T-1, ..., 1$ **do**
5:    $a \leftarrow \pi(s)$
6:    $s', d \leftarrow e.\text{step}(a)$
7:    **if** $d$ **then**
8:       $s \leftarrow e.\text{reset}()$
9:    **end if**
10:    $r \leftarrow \{\Psi^i\}_{i=1}^{N_c}(s)$
11:    $\mathcal{D}^{env} \leftarrow \mathcal{D}^{env} \cup \{s, a, r, s', d\}$
12:    **if** $t \% t_{cls\_int} == 0$ **then**
13:       **for** $t' = t_{cls\_steps}, t_{cls\_steps} - 1, ..., 1$ **do**
14:          postive samples $\mathcal{D}^{pos} \leftarrow \text{sample}(\mathcal{D}^{edit}, \mathcal{B})$
15:          recent negative samples $\mathcal{D}^{neg} \leftarrow \text{sample}(\mathcal{D}_{\lambda}^{env}, \mathcal{B})$
16:          $\mathcal{D} \leftarrow \text{mixup}((\mathcal{D}^{pos}, +1), (\mathcal{D}^{neg}, -1), \alpha)$
17:          train classifiers $\{\Psi^i\}_{i=1}^{N_c}$ on $\mathcal{D}$
18:       **end for**
19:    **end if**
20:    one-step training of $\pi, Q_1, Q_2$ follow Yarats et.al. [50]
21: **end for**

---

the push target is indicated with a green dot, which is at the bottom of the table. The reward of this environment consists of three parts: a dense reward for the reduction of L2 distance between the gripper and the cube, another dense reward for the reduction of L2 distance between the cube and the target, and a one-time reward of 10 points for successfully pushing the cube to the target.

**LED environment.** We develop a customized LED environment based on Robosuite, where the agent is expected to turn the color of the light from red to green. A cylinder indicating the LED light is placed at the bottom-right corner of the table, while a button is placed at the bottom-left corner of the table. Once the gripper touches the button or the LED light itself, the color of the LED will change from red to green and the task is considered finished. The reward of this environment consists of two parts: a dense reward corresponding to the reduction of L2 distance between the gripper and the button, and a completion reward of 10 points for successfully reaching the button or the cylinder.

### B.2 Hyper-parameters for goal image generation

We provide the hyper-parameters for appearance-based image editing tasks: Wipe (Sim), Wipe (Real), LED (Sim), and LED (Real) in Table 5, and the hyper-parameters for structure-based image editing tasks: Push (Sim) and Push (Real) in Table 6. For tasks that require the use of the feature extracting module in LfVoid, we provide the additional implementation details of DreamBooth in Table 7.

**Prompt tuning.** Prompt tuning is very lightweight for LfVoid as we simply visualize the attention maps and see which prompt results in an attention map that better associates each token with its region of interest. We only searched about 5 different prompts for LfVoid, and we also performed the same amount and type of prompt tuning for all the baselines.

**Optimization methods.** For the optimizations used during goal image generation, We stick to the exact optimization methods used in the released code of DreamBooth and Null-text Inversion: AdamW for DreamBooth and Adam for Null-text Inversion. The other parameters are all the default values provided by the official code.

| Task | Source Prompt | Target Prompt | Cross-Attention Editing Steps | Self-Attention Editing Steps | DreamBooth |
|---|---|---|---|---|---|
| Wipe (Sim) | A robot white table with markings on it | A robot white table with nothing on it | 40 | 0 | Yes |
| Wipe (Real) | A robot white table with markings on it | A robot white table with nothing on it | 40 | 0 | Yes |
| LED (Sim) | A red cylinder on a white table | A green cylinder on a white table | 40 | 50 | No |
| LED (Real) | A white table with a yellow duck light | A white table with a dark duck light | 40 | 0 | No |

Table 5: **Hyper-parameters for appearance-based visual goal generation.** The source and target prompt, as well as the editing steps are reported for appearance-based editing tasks. We also report the use of DreamBooth in the feature extracting module of LfVoid, please refer to Table 7 for details.

| Task | Prompt | Bounding-Box | Tokens | Editing Steps | DreamBooth |
|---|---|---|---|---|---|
| Push (Sim) | A photo of a sks cube and a gripper on a white table | [0.5, 0.7, 0.7, 0.9] | $\{5, 6, 15, ..., 44\}$ | 10 | Yes |
| Push (Real) | A photo of a sks cube and a gripper on a white table | [0.1, 0.3, 0.7, 0.9] | $\{5, 6, 15, ..., 49\}$ | 10 | Yes |

Table 6: **Hyper-parameters for structure-based visual goal generation.** The prompt, bounding box, token set, and the number of editing steps used in the P2P-DD algorithm. We also report the use of DreamBooth in the feature extracting module of LfVoid, please refer to Table 7 for details.

### B.3   Hyper-parameters for example-based RL

We provide the training details of the example-based RL algorithm in LfVoid in Table 8. Unless otherwise specified, we use the same hyperparameters as those in DrQv2. For each curve in the paper, we run 3 seeds and report the mean and 95% confidence interval of the results.

### B.4   Reward curve visualization

For the more challenging real-world scenarios, we visualize the reward function obtained with LfVoid on several successful trajectories. We first train the reward classifier using 60-80 raw observations as negative samples, and 60-80 edited goal images from these raw observations as positive samples. The training takes 200 epochs, with a batch size of 32 and a learning rate of 1e-4 using Adam as the optimizer. To evaluate these reward classifiers, we collect 5 trajectories for each task and use the trained classifiers to label each frame with a reward score. The trajectories have different lengths for different tasks because more steps are required to complete if the task is more complicated, such as the Wipe task where all the markings need to be cleaned. We also apply normalization on the obtained reward to let each reward curve start from zero at the first frame. We believe this practice enables a fair comparison between different reward functions, as it is the relative trend of the reward curve, rather than the absolute value, that is essential to the downstream RL training. We report the mean and the 95% confidence interval of the reward curve for each task over the collected trajectories.

| Task | Instance Number | Class Prompt | Class Images | Training Steps | Learning Rate |
|------|-----------------|--------------|--------------|----------------|---------------|
| Wipe (Sim) | 20 | a photo of a Franka robot gripper | 200 | 800 | 5e-6 |
| Wipe (Real) | 12 | a photo of a Franka robot gripper | 200 | 800 | 5e-6 |
| Push (Sim) | 11 | a photo of a red cube | 200 | 800 | 5e-6 |
| Push (Real) | 8 | a photo of a red cube | 200 | 800 | 5e-6 |

Table 7: **Hyper-parameters of the DreamBooth technique used in the feature extracting module of LfVoid.** The training details of DreamBooth for each task. DreamBooth extracts a special token that preserves the details of an object in the scene.

| Hyperparameter | Value |
|----------------|-------|
| Batch Size | 128 |
| Mixup alpha | 1 |
| Classifier Layer Sizes | [1024, 2] |
| Classifier Ensemble | 10 |
| Negative Sample Ratio $\alpha$ | 0.05 |
| Classifier Update Interval | 1000 |
| Classifier Update Steps | 10 |
| Edited Dataset Size | 1024 |
| Total Train Steps | 10e6 |

Table 8: **Example Based RL Hyper-Parameters.**

## C  Additional results

### C.1  Visual goal generation of LfVoid

We provide additional visual goal-generation results for each task in Figure 11 and Figure 12. These examples are selected from the datasets generated by LfVoid that are used to train example-based RL algorithms or reward functions for real-world evaluation.

### C.2  Visualization of RL with imagined goals

We also explored the line of work that uses VAEs to generate random, imagined goal frames and train a goal-conditioned policy to fulfill a user-provided goal image during test time. The generated image (randomly sampled from the latent space and decoded) is provided in Figure 13.

It can be observed that the images generated by VAEs are blurry and lose much of the details, which is unreliable for downstream example-based RL methods. This justifies the necessity of LfVoid to leverage large-scale pre-trained text-to-image models to provide plausible, high-quality goal images for successful visual RL. Moreover, the sampled goal images are very different from the actual user-specified goal states. This suggests that the random exploration policy can only encounter a small portion of the state space during training, and therefore, the goal-conditioned policy is not trained on the goal states similar to the ones during test time, which leads to unsuccessful results.

### C.3  Additional goal generation results of LfVoid in other environments

We demonstrate LfVoid's ability to generate realistic goal images in diverse environments by evaluating LfVoid on one simulated environment with obstacles and three real-world environments with a different robot arm (Ur5). The results are shown in Figure 14.

### C.4  LfVoid on general editing tasks

We show that the goal-generation method proposed by LfVoid is not limited to robotics environments and can be used to perform general-purpose editing. We include several general editing results in

Figure 15. The results show that LfVoid can perform more localized editing as well as preserve the background according to only text instructions when compared to existing editing methods. We only choose to study LfVoid 's performance in the robotics context because the ability to perform localized editing with high fidelity to the original image is important when using large-scale text-to-image models to guide robot learning.

## C.5 Human user evaluation

In Table 9, we present the user study results in terms of normalized Elo score. The evaluators show a strong preference for images generated by LfVoid compared to other methods. For the human user study on visual goal generation, We include 24 random subjects who do not have specialized knowledge in artificial intelligence. The subjects are shown images in Figure 4 and Figure 5, together with the editing instructions, and are then asked to provide rankings of each editing method based on how well the editing instruction is followed in each result. We also provide a snapshot of the questionnaire sent to the subjects in Figure 16.

We utilize the Glicko-2 rating system to analyze the ranking results. We first transform the rankings obtained in the questionnaire into comparisons: if method $m_1$ is ranked higher than method $m_2$, then $m_1$ is marked as a victory in the comparison between $m_1$ and $m_2$. Then we determine the rating of each method based on the initial score and the win-loss information contained in these comparisons, as suggested in the Glicko-2 system [53]. Finally, to standardize the scores within a scale from 0 to 100, we use the following formula to perform normalization:

$$R_{normal} = \frac{R - 800}{2000 - 800} * 100$$

|  | Imagic | InstructPix2Pix | NT-DD | NT-P2P-DD | Ours |
|---|---|---|---|---|---|
| Appearance-Based ($\uparrow$) | 47.7 | 42.4 | N/A | N/A | **87** |
| Structure-Based ($\uparrow$) | 43.6 | 18.2 | 58 | 63.8 | **91.6** |

Table 9: **User study on visual goal generation.** The Elo score shows user preference among multiple choices, higher is better.

## C.6 Visualization of reward functions

We also provide visualization videos of the reward functions obtained through LfVoid evaluated on both successful and failure trajectories. Please visit: `LfVoid.github.io`.

# D Limitations

Despite LfVoid's successful performance in editing images, it still has several limitations. We summarize the common failure cases into the following four categories: object deformations, background alterations, size changes, and incorrect associations. We will further discuss the causes of each failure case and suggestions to avoid these failures in the following paragraphs.

**Object deformations.** Objects might undergo deformations, including shape changes and color loss, between the input and edited images, as shown in Figure 17a. Hyper-parameter tuning can alleviate this problem by finding the best editing step for each input image, and the optimal value usually is the same for different input images of the same scene. But the optimal parameters for various scenes are usually different, thus requiring extra parameter tuning.

**Background alterations.** In some cases, details of the background scene may undergo unintended changes during the editing process, as shown in Figure 17b. The DreamBooth technique can mitigate this problem by fine-tuning the diffusion model on several images of the background to learn its details and structures. Performing grid-search on the hyper-parameter used in the editing techniques can also alleviate background alterations. In some cases, prompt engineering can also be a solution to this problem by adding additional tokens that describe the background to better preserve its details.

**Size changes.**    The use of Directed Diffusion may alter the size of the object, typically making it larger than its original size, as illustrated in Figure 17c. This phenomenon is the result of the imprecise control used in Directed Diffusion, which applies attention strengthening on the whole bounding-box area, rather than an exact subset of the area where we want the object to be placed. Moreover, the large-scale pre-trained diffusion models do not seem to possess the ability to change the shape of objects according to the viewpoint. Training diffusion models with knowledge of the physical laws may be needed to solve this problem.

**Incorrect associations.**    There have been instances where the model fails to associate objects with the corresponding prompt word, as shown in the examples in Figure 17d. Pre-trained diffusion models sometimes fail to correctly associate the input image with the given prompts, thus failing to follow the editing instruction. One solution to mitigate this problem is to use prompt engineering to select the best prompt that the model can "understand": the attention maps can precisely highlight the part of the image corresponding to each token. Training diffusion models with a better understanding of natural language and images may be the ultimate solution to this problem.

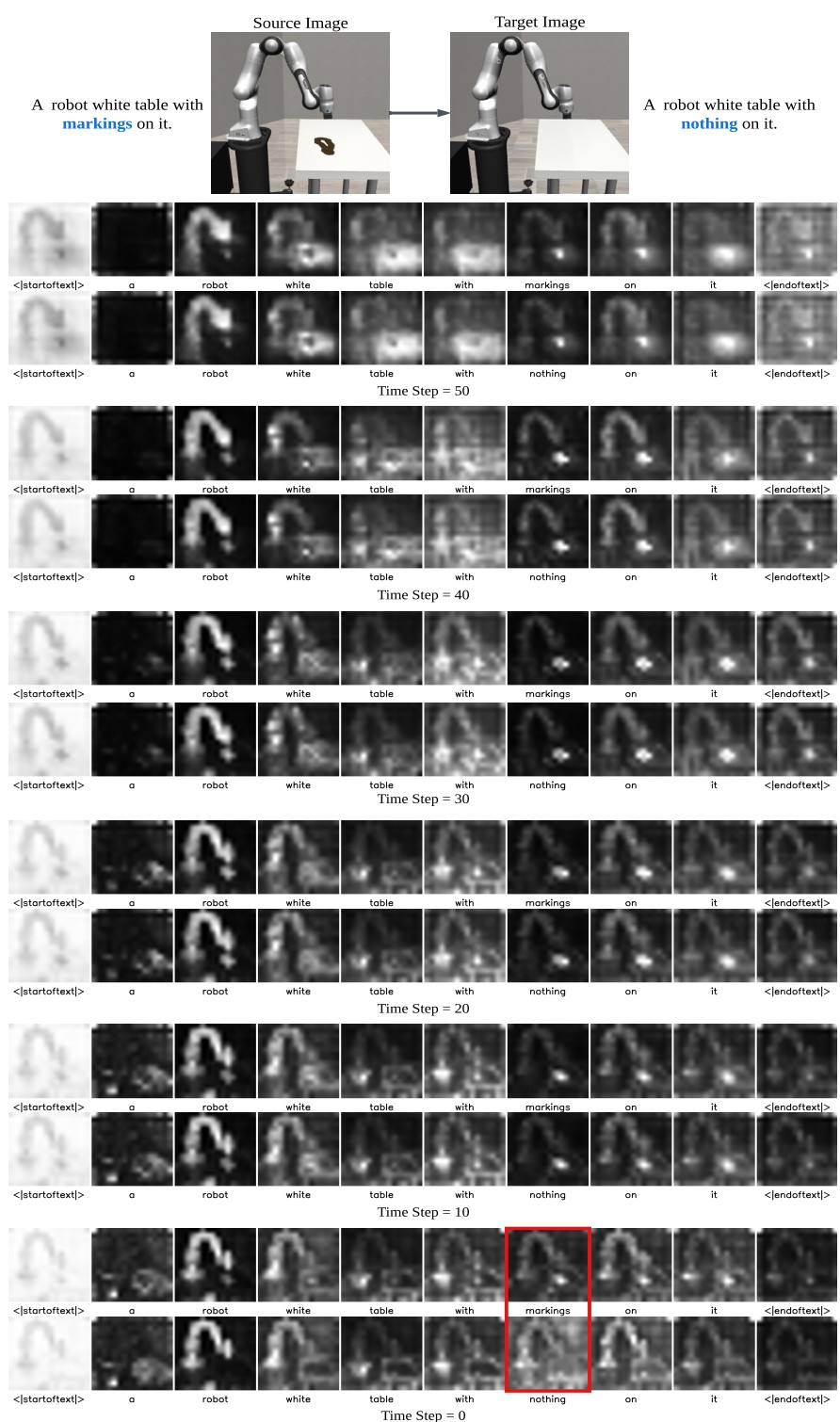

Figure 8: **Attention visualization of Prompt-to-Prompt.** The visualization of cross-attention maps at diffusion step $t = 50, 40, ..., 0$, with the cross-attention maps generated by the source diffusion process (upper row) and the target diffusion process (bottom row). The first 40 steps keep the original structure of the source image, and the required editing is performed during the last 10 diffusion steps, as indicated by the red box.

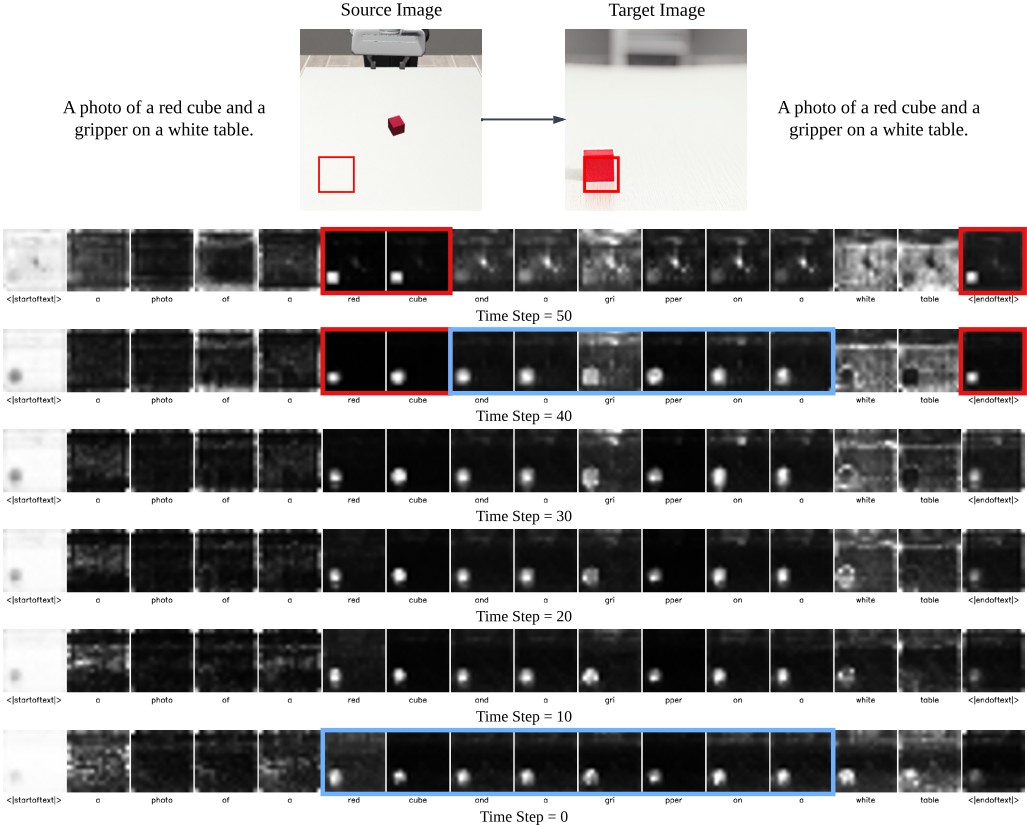

Figure 9: **Attention visualization of Directed Diffusion.** The visualization of cross-attention maps at diffusion step $t = 50, 40, ..., 0$. The attention strengthening and weakening suggested in Directed Diffusion are performed during the first 20 steps, as indicated by the red box. Although strengthening is initially applied on the bounding-box area, its effect will spread during the subsequent diffusion steps, leading to successful object displacement, as indicated by the blue box.

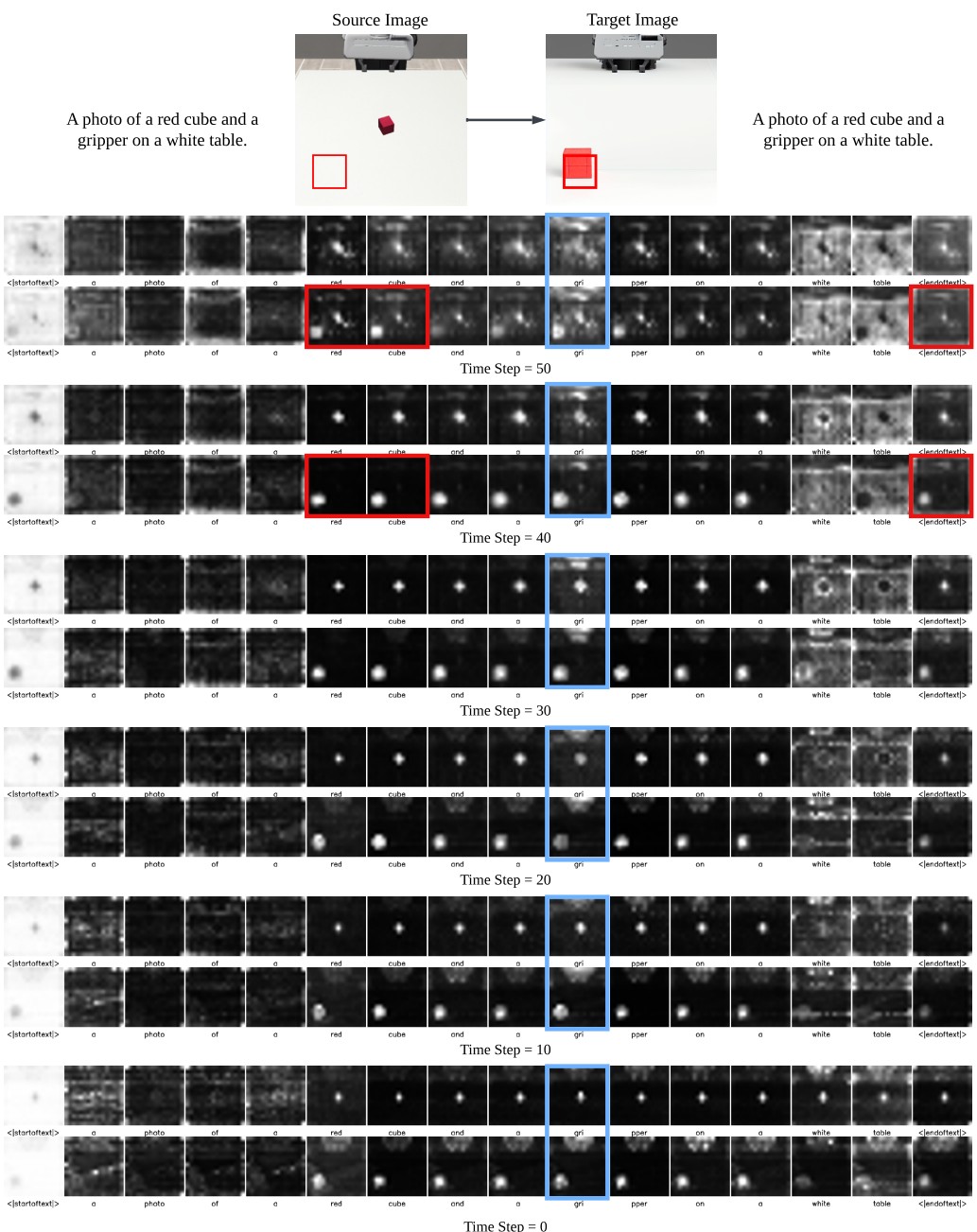

Figure 10: **Attention visualization of P2P-DD.** The visualization of cross-attention maps at diffusion step $t = 50, 40, ..., 0$. Attention map injection from source diffusion process (upper row) to target diffusion process (bottom row) is first performed at each step, preserving the background of the source image, as highlighted by the blue box. Then attention strengthening and weakening are applied on the target diffusion, as indicated by the red box, to generate an image that achieves the desired editing while preserving the original scene.

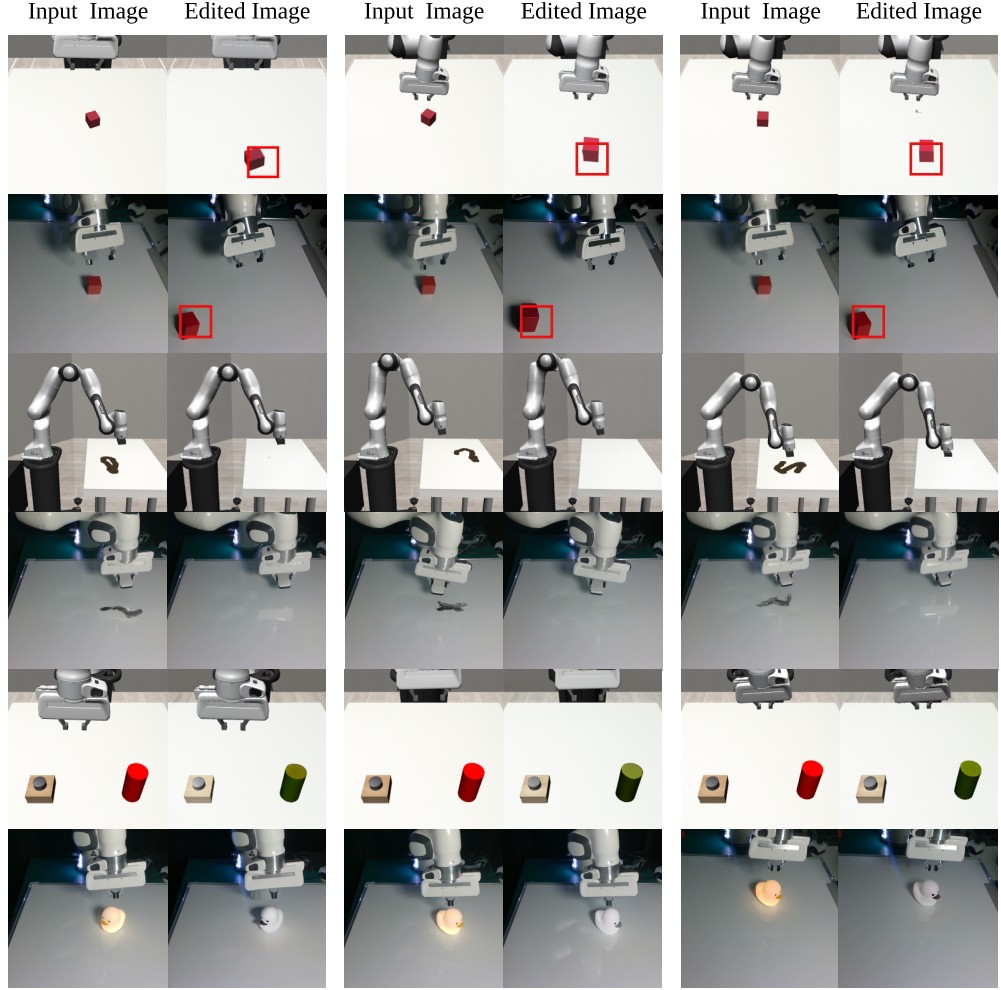

Figure 11: **Additional goal generation results.** We demonstrate several additional visual goal generation examples of LfVoid. From the top row to the bottom are the results of: Push (Sim), Push (Real), Wipe (Sim), Wipe (Real), LED (Sim), and LED (Real).

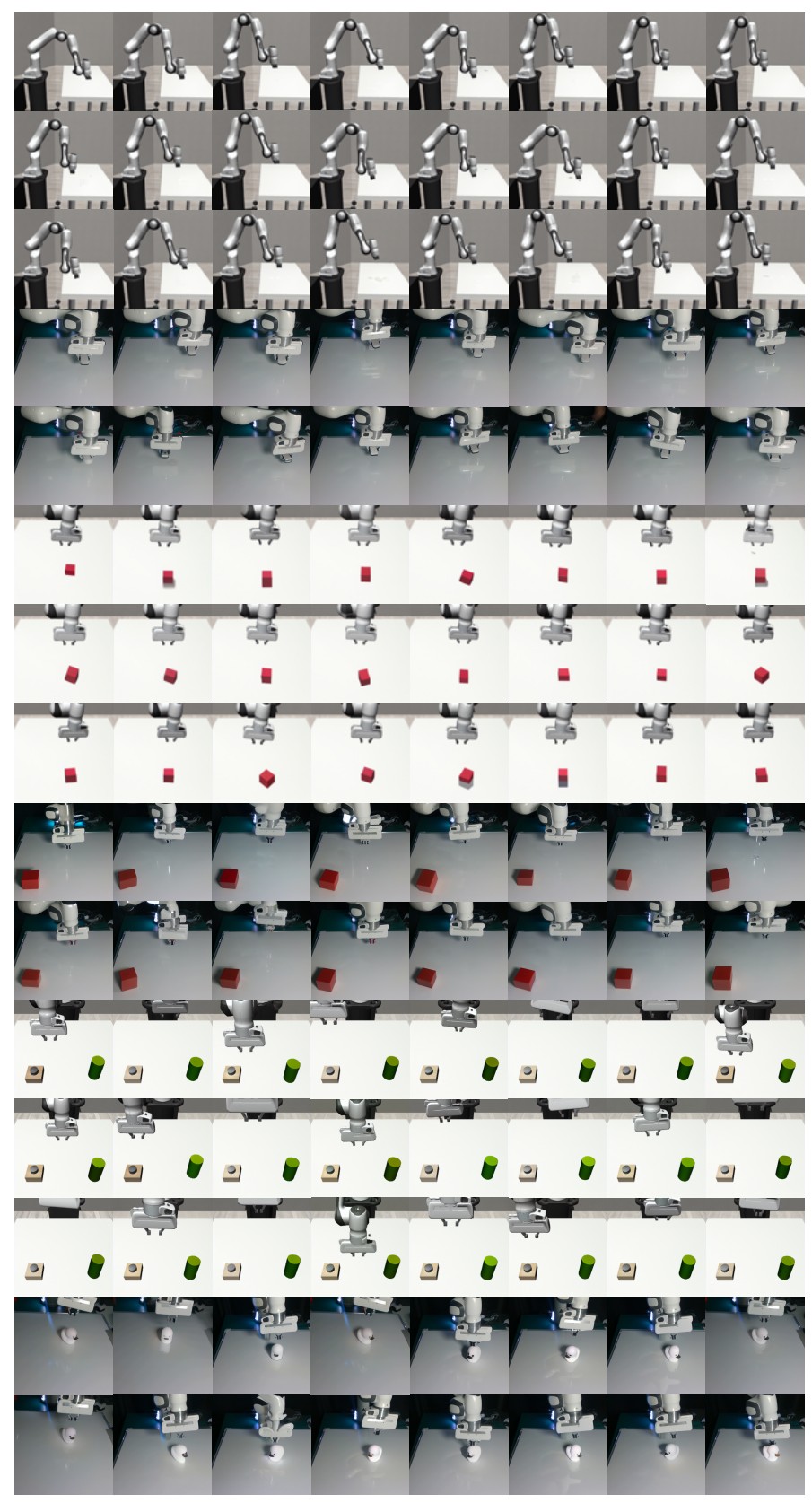

Figure 12: **Samples from the goal image dataset generated by LfVoid.**

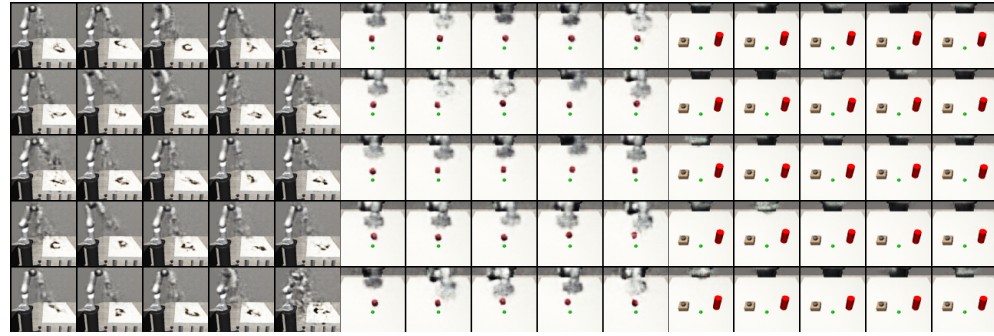

Figure 13: **Samples from the visual goal images from the representation space learned by visual RL with imagined goals methods.** We also explored the line of work that uses VAEs to randomly generate imagined goal frames and train a goal-conditioned policy on them. The imagined goals sampled from these methods are often too vague to be used for downstream example-based RL.

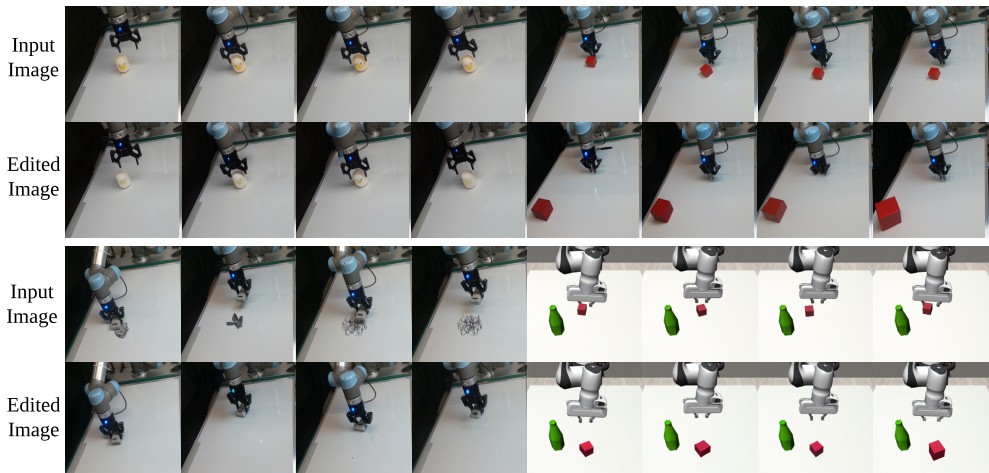

Figure 14: **Additional goal generation results of LfVoid on different real-world and simulation environment.** We test LfVoid's ability to generate goal images on three more real-world environments using an Ur5 robot arm, and one simulation task with an obstacle(a green bottle). LfVoid are able to perform successful editing in all four environments.

Input                LfVoid (Ours)              InstructPix2Pix              Imagic

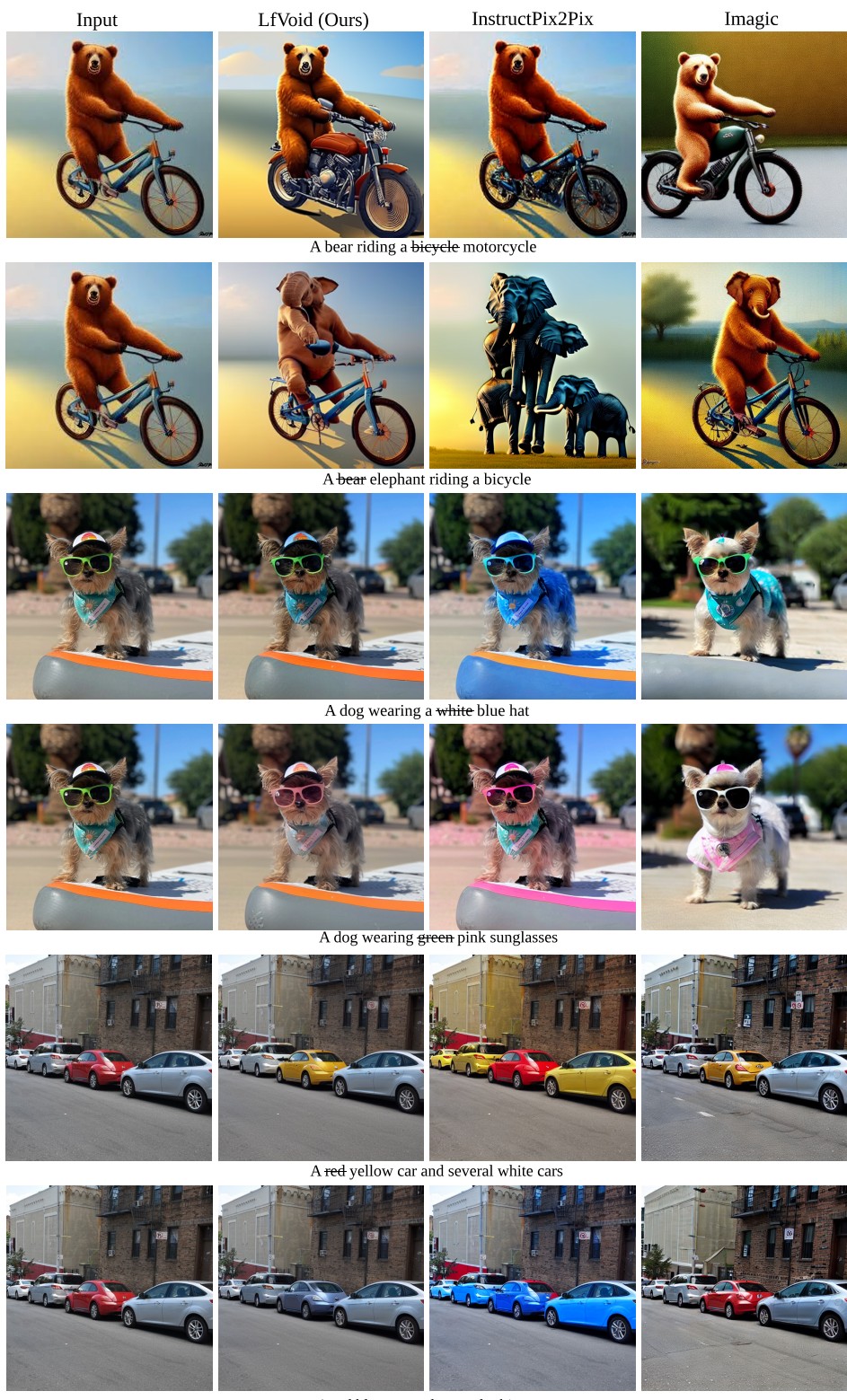

A bear riding a ~~bicycle~~ motorcycle

A ~~bear~~ elephant riding a bicycle

A dog wearing a ~~white~~ blue hat

A dog wearing ~~green~~ pink sunglasses

A ~~red~~ yellow car and several white cars

A ~~red~~ blue car and several white cars

Figure 15: **General editing results of LfVoid.** We show that LfVoid is a general editing method that can be used to perform general-purpose editing, with more localized editing and better preservation of the background.

**02** Which group of image better **clean the table** while preserving the rest of the scene? Please provide your rankings.

Original Image: A dirty table

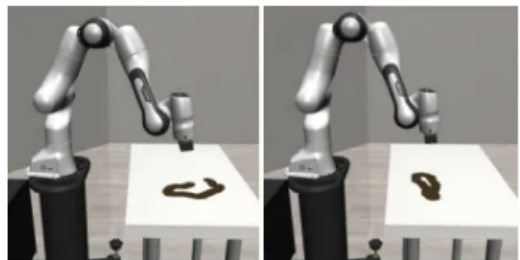

Please drag **the photos on the right side** to **the blank box on the left side** to provide your ranking.

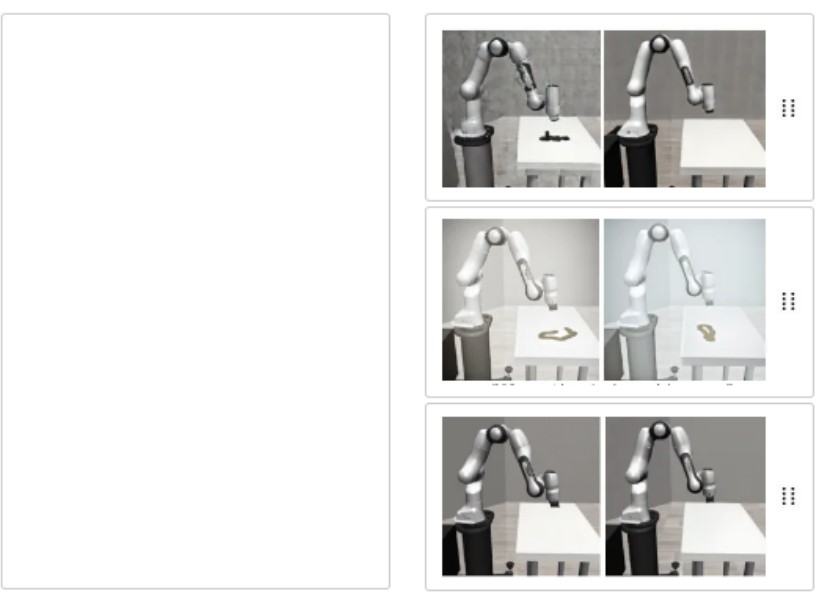

Figure 16: **A print screen for the questionnaire used in the user study.**

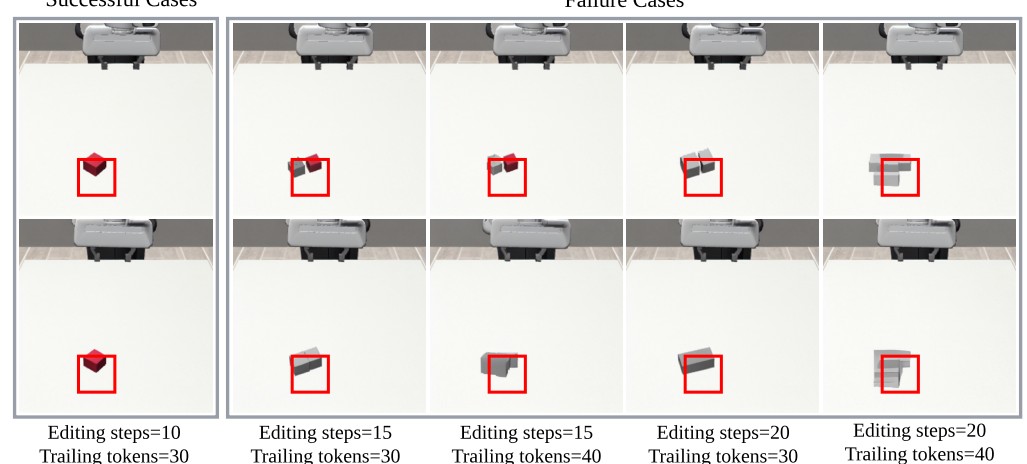

(a) **Failure cases: object deformations.** Different hyper-parameters may result in failure editing results that change the object's shape and color.

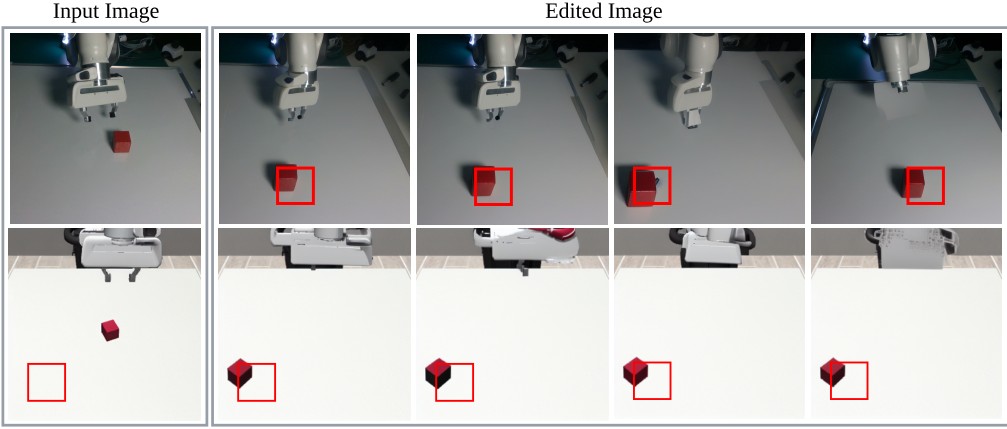

(b) **Failure cases: background alterations.** Details of the background may change during the editing process. The DreamBooth technique, performing grid-search on hyper-parameters, as well as prompt engineering can mitigate this problem.

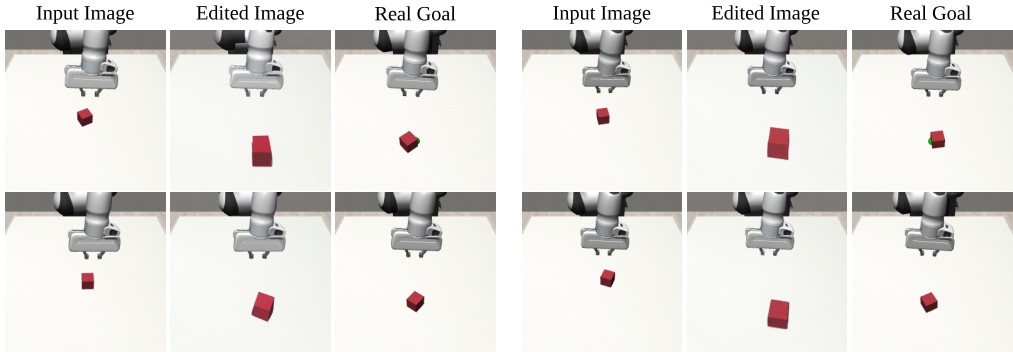

(c) **Failure cases: size changes.** The Directed Diffusion technique may cause the diffusion model to generate images with objects larger than the original ones.

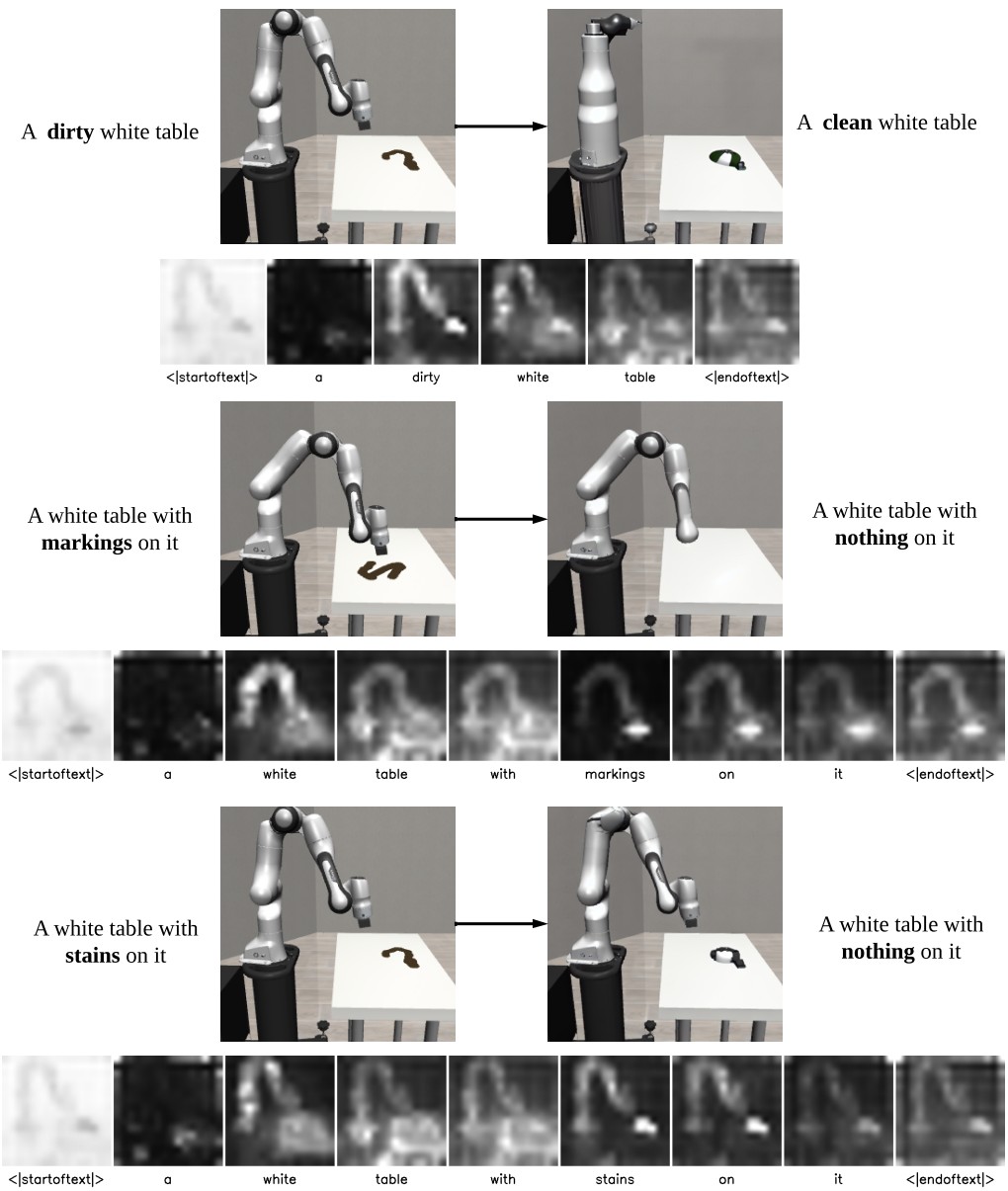

(d) **Failure cases: incorrect associations.** Different prompts may result in different editing qualities depending on how well the model can associate each token with its corresponding area in the input image.

Figure 17: **Failure cases of visual goal generation.** We demonstrate four common failure cases when using LfVoid to generate visual goals in various scenes: object deformations, background alterations, size changes, and incorrect associations.