# OpenReview forum: "Can Pre-Trained Text-to-Image Models Generate Visual Goals for Reinforcement Learning?"
_NeurIPS.cc/2023/Conference — NeurIPS 2023 poster_

### Official Review · Reviewer_WQFw · 2023-06-25

**Soundness:** 3 good
**Presentation:** 2 fair
**Contribution:** 1 poor
**Rating:** 5
**Confidence:** 4

**Summary:**

The objective of this paper is to harness the capabilities of pre-trained text-to-image models and image editing techniques in order to facilitate robot learning. This is achieved by leveraging these tools to modify the current scene towards the intended image objective.

**Strengths:**

The paper presents a novel approach that harnesses the knowledge embedded in large pre-trained generative models to facilitate zero-shot guidance for robot learning tasks. Additionally, the paper asserts that its edited image functionality outperforms existing methods. Lastly, the paper identifies a gap in current text-conditioned image generation models.

**Weaknesses:**

1. Figure 1 can be enhanced to effectively communicate the problem the author intends to address, while also succinctly highlighting their contribution and providing an overview of their proposed method.
2. In recent years, significant advancements in diffusion models have led to the emergence of notable works such as CACTI, GenAug, and ROSIE. These works have demonstrated considerable improvements in generative modeling and learning. However, none of these works were used as baselines in the paper, including DALL-E-Bot, which bears the closest resemblance to this work. While acknowledging technical differences, it would be valuable to compare and contrast the proposed method with these related works.
3. I appreciate the author's effort to compare various methods of image editing with their proposed approach. However, many of these methods are pretrained on different datasets, necessitating the evaluation of their performance on a comprehensive image editing dataset for a more equitable and meaningful comparison.
4. The author's inclusion of both simulation and real-world demonstrations is commendable. However, to enhance the persuasiveness of the findings, it would be beneficial to introduce slightly more complex tasks or incorporate additional distractors during evaluation to assess the robustness of the proposed method.
5. The author may also consider exploring the line of work that involves generating demonstrations using video editing techniques for robot learning, as exemplified in "Human-to-Robot Imitation in the Wild," RSS 2022.

**Questions:**

I hope the authors can address all the questions and concerns i have in the weakness section.

**Limitations:**

Yes, the authors have addressed all the limitations.

---

> ### Author Rebuttal · Authors · 2023-08-10
>
> Dear **Reviewer WQFw**, we thank you for your detailed and thorough review. In the following sections, we seek to address each of your concerns.
>
> ---
>
> **Q**: Figure 1 can be enhanced to effectively communicate the problem the author intends to address, while also succinctly highlighting their contribution and providing an overview of their proposed method.
>
> **A**: We thank the reviewer for this suggestion and the updated Figure 1 is shown in Figure 5 of the attachment. We added a subfigure picturing what the problem is that we are trying to address.
>
> **Q**: In recent years, significant advancements in diffusion models have led to the emergence of notable works such as CACTI, GenAug, and ROSIE. These works have demonstrated considerable improvements in generative modeling and learning. However, none of these works were used as baselines in the paper, including DALL-E-Bot, which bears the closest resemblance to this work. While acknowledging technical differences, it would be valuable to compare and contrast the proposed method with these related works.
>
> **A**: CACTI, GenAug, and ROSIE use diffusion models to perform data augmentation on existing expert demonstrations, improving the robustness and generalization ability of the manipulation policy. It should first be noted that LfVoid aims to solve a different problem when compared to this line of work: we utilize the large-scale diffusion model to translate language instructions into corresponding goal images. Therefore, the outcome of solving the task is not provided by the demonstrations but by the large-scale pre-trained model. That being said, the distinction between LfVoid and previous works is clear. While acknowledging the differences, the use of diffusion models to perform image editing is similar in these works and LfVoid. The editing modules in this line of work mainly perform localized image inpainting conditioned on human-specified or automatically generated object masks. Since the models used in these works are not open-sourced, we cannot directly compare LfVoid with them.
>
> DALL-E-Bot is a task-specific method that can only solve object rearrangement tasks, and thus its policy cannot generalize to our manipulation tasks. However, we are able to compare the goal generation method proposed by DALL-E-Bot with LfVoid. More specifically, we provide human-specified masks to DALL-E 2 model and the text prompt describing the edited image, and report the qualitative results in Figure 1 and Figure 2 and the quantitative results in Table 1 and Table 2 of the attachment. Despite DALLE-2 having the advantage of a user-specified mask (the region outside the mask will remain unchanged), LfVoid still has better or comparable performance in all the tasks.
>
>
> **Q**: I appreciate the author's effort to compare various methods of image editing with their proposed approach. However, many of these methods are pre-trained on different datasets, necessitating the evaluation of their performance on a comprehensive image editing dataset for a more equitable and meaningful comparison.
>
> **A**: We would like to point out first that both LfVoid and Imagic are **fine-tuning** methods, not models, that can be used on any pre-trained diffusion models. To ensure a fair comparison, we use the same open-source StableDiffusion model(v1-4) in our experiments when evaluating LfVoid and Imagic.
>
> Secondly, while Instruct-pix2pix is indeed a released model, it also builds upon the open-source StableDiffusion model: it fine-tunes the model parameters with generated paired data so that the model can perform editing based on an input image and a text instruction.
>
> Therefore, we believe that our comparison between these methods is reasonable since all three editing methods build upon the same pre-trained StableDiffusion Model.
>
>
> **Q**: The author's inclusion of both simulation and real-world demonstrations is commendable. However, to enhance the persuasiveness of the findings, it would be beneficial to introduce slightly more complex tasks or incorporate additional distractors during evaluation to assess the robustness of the proposed method.
>
> **A**: We have added a distractor object to push-sim, and have introduced three real-world tasks using the UR5 robot arm and reported the goal generation results in Figure 3 in the attachment.
>
> **Q**: The author may also consider exploring the line of work that involves generating demonstrations using video editing techniques for robot learning, as exemplified in "Human-to-Robot Imitation in the Wild," RSS 2022.
>
> **A**: LfVoid uses image editing methods to generate the goal frame for RL learning, while WHIRL uses video inpainting methods to align human interaction videos and robot interaction videos, and therefore can measure how good the robot performs compared with target human videos. The editing methods create a representation space for this objective function which selects the best robot interactions to perform imitation learning. The comparison with this line of work is very interesting and we will add this line of work to the references.
>
> We kindly refer reviewers to the general response section for common questions. If there's anything unclear in the above, we're more than glad to further discuss the details.

---

> > ### Comment · Reviewer_WQFw · 2023-08-12
> > **Thank you**
> >
> > I would like to thank the authors for the clarification and response.

---

> > > ### Author Response · Authors · 2023-08-13
> > > **Further Discussion**
> > >
> > > Dear reviewer WQFw,
> > >
> > > Thank you for acknowledging our clarification and response.
> > >
> > > Should there remain any unresolved concerns of the paper you feel need further explanation, please let us know. We are eager and open to further discussions to ensure clarity and comprehension.
> > >
> > > If you find that our responses have adequately addressed the initial concerns, we kindly request you to consider revisiting the rating of the paper. Your insights and evaluation are crucial to us, and we sincerely hope our efforts align with your expectations.

---

### Official Review · Reviewer_NX6j · 2023-07-03

**Soundness:** 2 fair
**Presentation:** 2 fair
**Contribution:** 2 fair
**Rating:** 5
**Confidence:** 2

**Summary:**

Post rebuttal updating score to 5

This work proposes to use generated visual goals for RL. A diffusion model based approach is used to edit visual observations based on text prompts. The proposed image editing approach is shown to be better than prior text-based image editing approaches based on a human evaluation. Visual goals predicted using the proposed method shows improved RL performance compared to baselines that use other image editing mechanisms.

**Strengths:**

* Leveraging pre-trained models for image editing is an interesting idea
* Proposed approach seems to work better than some prior editing approaches
* Some indication that the generated visual goals help learning


**Weaknesses:**

Originality: Imagining visual goals for RL has been explored in prior work (e.g., [1] https://arxiv.org/pdf/1807.04742.pdf), which limits the originality of this work. The use of diffusion models could be new.

Significance:
* Experimental results do not demonstrate the need for visual goals. For instance, there are no baselines in the RL experiments which are not based on imagined goals.
* Very few tasks are tested
* Baseilnes are inadequate
* Approach is too specific to the tasks considered and generality of approach is questionable

Presentation/Clarity
* Need better clarity on motivation, background on methods, problem formulation and approach overview
* There needs to be a clear motivation/overview section before section 3. Technical details lack context and are difficult to follow. For instance, section 3 talks about latent diffusion models without providing any background/motivation. There needs to be a technical overview section on diffusion models.
* Method description is vague and mathematical details are missing.
  - 3.1.1 Description of the optimization approach is vague.
  - 3.1.2 What is inversion and why is it necessary?
  - 3.1.3 Assumes prior knowledge and familiarity with diffusion models. What is x_0? What does it mean to replace attention maps?

Experiments
* Is the comparison on image editing fair? The proposed method is fairly specific to the tasks considered while it is compared against general purpose image generation/editing techniques such as pix2pix.
* Inadequate baselines for the RL experiment: There seem to be no baselines that are not based on imagined goals/image editing.


**Questions:**

* Many details about the setting are unclear. Need better clarity on the following.
  - Where do the text prompts/instructions for editing come from?
  - What are the inputs and the outputs of the components of the approach?
  - How does learning work?
* Is it possible to use any quantitative metrics for image editing experiments (e.g., based on CLIP encoders)?
* For the manually defined visual goals how many images were manually created? How many task instances are there altogether?
* line 297 mentions ‘a certain level of prompt tuning is needed’, but such details are missing from the main text.


**Limitations:**

See above

---

> ### Author Rebuttal · Authors · 2023-08-10
>
> Dear **Reviewer NX6j**, we thank you for your detailed and thorough review. In the following sections, we seek to address each of your concerns.
>
> ---
>
> **Q**: Experimental results do not demonstrate the need for visual goals: no baseline in the RL experiments that are not based on imagined goals.
>
> **A**: We would like to point out that the CLIP baseline mentioned in line 260 is not based on imagined goals. It only requires a user-specified text prompt and the CLIP distance between the text prompt and observation image is used as a reward to train RL. Results show that LfVoid clearly outperforms it, demonstrating the advantage of providing visual guidance.
>
> **Q**: There needs to be a clear motivation/overview section before section 3. Technical details lack context and are difficult to follow.
>
> **A**: We appreciate your advice and will polish the paper to improve the delivery in the final version. Particularly, we will improve the corresponding part of the paper. Due to the length limit, it's hard to go into too much detail about the technical backgrounds of the diffusion model, and it's common practice to refer the readers to relevant papers for more detail, including the image editing baselines used in LfVoid [1,2,3], nevertheless, we agree that it's always better to provide sufficient background in the paper itself.
>
> **Q**: Descriptions of the optimization approach is vague.
>
> **A**: For the optimization method used in goal generation, we use the exact optimization methods used in the released code of DreamBooth and Null-text Inversion: AdamW for DreamBooth and Adam for Null-text Inversion. The other parameters are all the default values provided by the official code.
>
> **Q**: What is inversion and why is it necessary?
>
> **A**: We need to invert the provided source image to a diffusion process so that our editing module can generate the target image conditioning on both the source image and the text prompt. Diffusion models could only synthesize a target image conditioning on a text prompt. Through inversion, we can extrinsically condition the generation process on both the source image and the text prompt[3]. We provide results where we eliminate the inversion module completely: Dreambooth P2P in Figure 1 and Dreambooth P2P-DD Figure 2 in the attachment, and observe that the generated target image differs greatly from the source image.
>
> **Q**: What is x_0? What does it mean to replace attention maps?
>
> **A**: $x_0$ denotes the final generated image at the last diffusion step. During each generation step, the diffusion model uses cross-attention to process the image and the text prompt, and attention maps are produced. We can replace the attention maps between the diffusion processes that separately generate the source and target image. Please refer to line 130 and the original paper[1] for more details.
>
> **Q**: Where do the text prompts/instructions for editing come from?
>
> **A**: They are provided by humans, which is a description of the desired goal image.
>
> **Q**: What are the inputs and outputs of the components of the approach?
>
> **A**:  For the Feature Extracting Module, the input is several initial visual observations and a prompt describing the object we want to remember. The output is a fine-tuned model using the <sks> token to represent that object.
>
> For the Inversion Module, the input is the source image and the text prompt description. The text prompt can contain the <sks> token to better describe the scene. The output is a noise vector and a series of optimized null-text embeddings for each diffusion timestep.
>
> For the Editing Module, the input includes both the input and output of the Inversion Module, as well as the text prompt describing the target image(for appearance-based editing), or the bounding box and trailing tokens(for structure-based editing). The output is the edited image.
>
> We will include this in Section 3.1.
>
> **Q**: How does learning work?
>
> **A**: We assume this is asking about the procedure of the RL. Generally speaking, it consists of two iterative procedures. Given a set of generated goal images, a classifier is trained to distinguish the goal image and observation encountered by an RL agent from a replay buffer. Meanwhile, the RL agent is optimized to gain more reward from the classifier to achieve the states that assemble the goal image, granting it the ability to reach the desired goal during test time.
>
> **Q**: For the manually defined visual goals how many images were manually created?
>
> **A**: In the pipeline of LfVoid, there are no manually defined visual goals since LfVoid can synthesize the visual goals requiring only a user-defined text prompt and/or bounding box. For evaluation purposes, we sample 1024 visual goals to test the upper bound of LfVoid and report this result as a baseline.
>
> **Q**: Unfair comparison to general purpose methods.
>
> **A**: We would like to argue that the goal-generation method is a general editing method that can perform various editing tasks that are not limit to robotics context and therefore the comparison on image editing is fair. Secondly, the example-based learning of LfVoid is also a general-purpose method compatible with any visual RL setting. The design choices and algorithms of LfVoid are not specific to those tasks.
>
> We kindly refer reviewers to the general response section for common questions. If there's anything unclear in the above, we're more than glad to further discuss the details.
>
> [1] Hertz, Amir, et al. "Prompt-to-prompt image editing with cross attention control." arXiv preprint arXiv:2208.01626 (2022).
>
> [2] Kawar, Bahjat, et al. "Imagic: Text-based real image editing with diffusion models." Proceedings of the IEEE/CVF Conference on Computer Vision and Pattern Recognition. 2023.
>
> [3] Mokady, Ron, et al. "Null-text inversion for editing real images using guided diffusion models." Proceedings of the IEEE/CVF Conference on Computer Vision and Pattern Recognition. 2023.

---

> > ### Comment · Reviewer_NX6j · 2023-08-15
> > **Thank you for the response**
> >
> > I appreciate the response.
> >
> > Clarity issues were partially addressed by the author response. I strongly recommend improving the clarity of the draft to make it more accessible to a wider audience.

---

> > > ### Author Response · Authors · 2023-08-17
> > > **Thank you**
> > >
> > > We thank the reviewer for the effort and the positive feedback. We will continue to polish the paper in the final version.

---

### Official Review · Reviewer_pQmz · 2023-07-03

**Soundness:** 3 good
**Presentation:** 4 excellent
**Contribution:** 4 excellent
**Rating:** 6
**Confidence:** 3

**Summary:**

This paper demonstrates a novel way to utilize existing large-scale text-to-image models for robot learning. Specifically, they modify existing pre-trained text-to-image models to produce visual goals for example-based reinforcement learning, before learning policies from the generated images. Experimentation in both simulation and the real world shows that the proposed modifications to pre-trained text-to-image models are able to generate higher-fidelity images than the baselines, and that this also leads to gains in downstream performance.

**Strengths:**

- Well-written. The paper is well-written, and the figures aid in the understanding of the methodology.
- Novelty. The proposed approach seems to be a novel one which builds on ideas from the literature.
- Real world experiments. The inclusion of real world experiments for both the image goal generation and the RL training make the strength of the proposed approach a lot more convincing than otherwise.
- Strong results. The proposed approach outperforms baselines from prior work as well as the ablations. The qualitative results provided in Figures 4 and 5 display impressive performance of the model.

**Weaknesses:**

- Evaluation. While both qualitative and quantitative metrics for image goal generation are included, the worry is that the qualitative examples may be cherry-picked and that the user-study done for a quantitative assessment may not be extensive enough. Would it be possible to use the standard metrics which conditional image generation papers utilize to quantitatively evaluate the proposed approach? For instance, once the ideal goal image has been generated, could the output of the proposed method be compared to using either MSE or LPIPS [1]? Furthermore, would it be possible to provide more qualitative examples? There are about three more examples per task in the appendix, but these all seem to be very similar to one another and limited in diversity. Particularly, more visualizations of the real world images would be appealing as this setting is much harder than the simulated images.

- Prompt tuning. Ideally, at test-time, you are just given the current image, and a text-based goal prompt, nothing else. Line 296 says that "a certain level of prompt tuning is needed in order to achieve optimal editing performance." How much prompt tuning was done for the method? Was the same done for the baselines?

[1] Richard Zhang, Phillip Isola, Alexei A Efros, Eli Shechtman, and Oliver Wang. The unreasonable effectiveness of deep features as a perceptual metric. In IEEE Conf. Comput. Vis. Pattern Recog., 2018.


**Questions:**

- Qualitative Results. While the language instructions for Imagic and the proposed method are the same, ensuring a fair comparison, why are they different for InstructPix2Pix? Furthermore, why is the ablation only done for Figure 5, and not for Figure 4?

- RL Experiments. Were all hyper-parameters in the RL training the same across methods? In Figure 7, the number of trajectory steps seems to vary across tasks -- why is this the case? Are the trends different if more trajectory steps were allowed?

**Limitations:**

Yes, the authors have adequately addressed the limitations of the work.

---

> ### Author Rebuttal · Authors · 2023-08-10
>
> Dear **Reviewer pQmz**, we thank you for your detailed and thorough review. In the following sections, we seek to address each of your concerns.
>
> ---
>
> **Q**: Would it be possible to provide more qualitative examples? There are about three more examples per task in the appendix, but these all seem to be very similar to one another and limited in diversity.
>
> **A**: We have provided more diverse qualitative results of LfVoid in Figure 7 of the attachment. Moreover, we add three additional real-world environments with a UR5 robot arm and one simulation task with a distractor to further increase the diversity. The results are shown in Figure 3 of the attachment and we observe that LfVoid can successfully perform all the desired edits in various environments.
>
>
> **Q**: Why is the ablation only done for Figure 5, and not for Figure 4?
>
> **A**: We thank the reviewer for this question and have added additional ablations for both Figure 5 and Figure 4. Please refer to Figure 1 and Figure 2 in the attachment. It should be noted that LED(Sim) and LED(Real) can perform the desired editing without the DreamBooth token and therefore we did not use DreamBooth when generating the goal images of these two environments.
>
>
> **Q**: While the language instructions for Imagic and the proposed method are the same, ensuring a fair comparison, why are they different for InstructPix2Pix?
>
> **A**: Both LfVoid and Imagic expect the text prompt to directly describe the goal image, while InstructPix2Pix expects the text prompt to describe the editing instructions to perform on the input image. This results in a slightly different prompt for these methods.
>
> **Q**: Were all hyper-parameters in the RL training the same across methods?
>
> **A**: Yes, the hyper-parameters are the same and reported in Table 6 in the Appendix.
>
>
> **Q**: In Figure 7, the number of trajectory steps seems to vary across tasks -- why is this the case? Are the trends different if more trajectory steps were allowed?
>
> **A**: This is because different tasks require different numbers of steps to finish when humans are collecting the demonstrations. For Wipe-Real, we found that it needs about 18 steps to completely wipe off the markings. And for LED-Real the number is 16, for Push-Real the number is 14.
>
>
> We kindly refer reviewers to the general response section for common questions. If there's anything unclear in the above, we're more than glad to further discuss the details.

---

> > ### Comment · Reviewer_pQmz · 2023-08-21
> >
> > Thanks for the response. The authors' clarifications have addressed all of my questions.

---

> > > ### Author Response · Authors · 2023-08-22
> > > **Thank you**
> > >
> > > We thank the reviewer for the feedback.

---

### Official Review · Reviewer_jYPU · 2023-07-06

**Soundness:** 3 good
**Presentation:** 4 excellent
**Contribution:** 3 good
**Rating:** 7
**Confidence:** 4

**Summary:**

In this paper, authors propose a new method called LfVoid to tackle RL tasks. LfVoid can generate consistent visual goal frames through using pretrained LDM and subsequently train a discriminator to output rewards for downstream RL task. In order to improve editing consistency, DreamBooth, null-text inversion, P2P and Directed diffusion are incorporated into the pipeline.

**Strengths:**

1. This paper is well written, clear and easy to understand.
2. An organic integration of several techiniques for controlled generation
3. Compared with existing editing methods, only the proposed LfVoid correctly modifies the desired objects and makes no obvious changes to other irrelevant things.
4. The improved generative quality subsequently enhance the performance of example-based RL.

**Weaknesses:**

1. LDM is kind of overkilling in this benchmark.
2. Lack comparison with other RL methods leveraging goal frame generation

**Questions:**

1. The evaluation benchmark is a little simple which more or less wastes the strong expressive power of pretrained LDM. What is the advantage of using LDM in your application compared with training a latent VAE to generate the goal frame except for less training frames?
2. As mentioned in summary, 4 techniques, including DreamBooth, null-text inversion, P2P and Directed diffusion, are incorporated but the ablation study only compares null-text P2P-DD and null-text P2P. In my opinion, the effects of DreamBooth and null-text inversion are overlapped. Can you provide any experiment to validate the improvement brought by DreamBooth token? Also, I understand the difficulty of ablation study in this work due to the lack of effective quantitative comparison. Is it possible to design some new metrics that only focus on the region of interest since the bounding boxes are provided?
3. Comparison with existing RL methods leveraging goal frame generation, for example, Goal-Aware Prediction[1], was not provided. Can you compare LfVoid  with these baselines or explain why this is not meaningful?

[1]. Nair, S., Savarese, S. &amp; Finn, C.. (2020). Goal-Aware Prediction: Learning to Model What Matters. <i>Proceedings of the 37th International Conference on Machine Learning</i>, in <i>Proceedings of Machine Learning Research</i> 119:7207-7219 Available from https://proceedings.mlr.press/v119/nair20a.html.



**Limitations:**

This paper solved the image editing problem required for goal frame generation. A solid contribution was made. I would increase my rating if my concerns are properly addressed.

---

> ### Author Rebuttal · Authors · 2023-08-10
>
> Dear **Reviewer jYPU**, we thank you for your detailed and thorough review. In the following section, we seek to address each of your concerns.
>
> ---
>
> **Q**: "LDM is kind of overkilling in this benchmark." and the advantage of LDM over VAE to generate the goal frame.
>
> **A**: In LfVoid, the LDM bears the task of **editing** the current observation based on a **text description**, while keeping the irrelevant parts of the image **unchanged**. While there exist works like [1] that utilize CVAEs to generate text-conditioned images or use VAE along with GAN for text-conditioned image editing [2], their ability to fulfill all three requirements of LfVoid is limited. To the best of our knowledge, the pipeline proposed by LfVoid that combines LDMs with inversion, attention control, and object specification is the only method that makes the downstream example-based RL procedure possible. We attribute this to large LDM models’ better understanding of the semantic meaning of the world.
>
> We also explored the line of work that uses VAEs to generate random frames and train a goal-conditioned policy to fulfill a user-provided goal image during test time. The generated image (randomly sampled from the latent space and decoded) can be seen in Figure 6 of the attachment, it's clear that those images are blurry and lose much of the details, which is unbearable for downstream example-based RL methods.
>
> **Q**: "Lack comparison with other RL methods leveraging goal frame generation, for example, Goal-Aware Prediction[3], was not provided."
>
> **A**: Thank you for your advice. This is indeed reverant to our method and we do think the comparison with such methods is valuable. We have conducted two additional experiments, Goal-Aware Prediction(GAP) and its predecessor work Visual Reinforcement Learning from Imaginary Goals (VIG)[4].
>
> GAP first collects transitions from the environment using a random policy, then trains a dynamic model in latent space on these data. During test time, GAP uses model predictive control to achieve a **user-provided** goal image. VIG collects a random dataset as well and trains a VAE over the image observations, then trains a goal-conditioned RL agent to fulfill sampled goals from the VAE.
>
> We want to point out that the goal image (at test time) in both settings is user-provided, which means the user has to figure out one way to achieve the goal state before ever having a feasible policy to achieve it. What's more, since both these method uses random policy to collect data, it's hard for them to have a faithful knowledge of the dynamics or distribution over the goal image since it can be difficult for the random policy to ever reach such a goal state (like wipe all the markings off the table)
>
> In the additional results, we provide an oracle goal image (as contrust to the edited images from LfVoid) to both methods to make them fulfill the goal state. The rewards of the three simulated tasks can be found in Table 5 in the attachment. It's clear that they can not achieve any of the tasks despite providing the oracle goal observation.
>
>
> **Q**: "As mentioned in summary, 4 techniques, including DreamBooth, null-text inversion, P2P and Directed diffusion, are incorporated but the ablation study only compares null-text P2P-DD and null-text P2P"..., "Can you provide any experiment to validate the improvement brought by DreamBooth token?"
>
> **A**: We thank the reviewer for this question and have provided more ablation studies in Figure 1 and Figure 2 of the attachment. In particular, we report the results of removing the DreamBooth module (Null-text P2P and Null-text P2P-DD) and observe that the DreamBooth token contributes significantly to preserving the background details and performing the desired edit. Additionally, we report quantitative ablation results in Table 3 and Table 4 of the attachment.
>
> We kindly refer reviewers to the general response section for common questions. If there's anything unclear in the above, we're more than glad to further discuss the details.
>
> [1] Zhang, Chenrui, and Yuxin Peng. "Stacking VAE and GAN for context-aware text-to-image generation." 2018 IEEE Fourth International Conference on Multimedia Big Data (BigMM). IEEE, 2018.
>
> [2] Pernuš, Martin, et al. "Fice: Text-conditioned fashion image editing with guided gan inversion." arXiv preprint arXiv:2301.02110 (2023).
>
> [3] Nair, Ashvin V., et al. "Visual reinforcement learning with imagined goals." Advances in neural information processing systems 31 (2018).
>
> [4] Nair, Suraj, Silvio Savarese, and Chelsea Finn. "Goal-aware prediction: Learning to model what matters." International Conference on Machine Learning. PMLR, 2020.

---

> > ### Comment · Reviewer_jYPU · 2023-08-17
> > **Response to rebuttal**
> >
> > Most of my concerns are well addressed. Therefore, I am happy to increase my rating to accept.

---

> > > ### Author Response · Authors · 2023-08-17
> > > **Thank you**
> > >
> > > We thank the reviewer for the effort and the positive feedback.

---

### Author Rebuttal · Authors · 2023-08-10

Dear reviewers, we appreciate all your helpful feedback. In this response, we address the common questions and comments. We welcome further discussion with each reviewer to address any remaining concerns.

We’d like to thank Reviewer jYPU for acknowledging that our method is “An organic integration of several techniques for controlled generation” that “subsequently enhance the performance of example-based RL”, and Reviewer pQmz for acknowledging our work is "A lot more convincing" since the introduction of real world experiments and demonstrates "impressive performance".

Our work (LfVoid) explored the possibility of utilizing the intrinsic knowledge embedded in pertained latent diffusion models (LDM) to provide visual guidance for robotic reinforcement learning. Starting from the current observation and language instructions on the desired goal, LfVoid trains an RL agent to achieve the desired goal with **zero need** for **expert demonstrations** or human-designed **reward functions**.

In the following part, we address some of the common concerns of the reviewers:

**Quantitative evaluation over generated images**

We thank the reviewers for this suggestion. We have reported the LIPIS distance and L2 distance between the edited images and real goal images in Table 1 and Table 2 of the attachment. We also report several more ablation tasks to make the ablation study more concrete. It's clear from the results that LfVoid is not only superior to the other baselines but also better with all its components combined.

**The ability of LfVoid for general-purpose image editing**

We show that the goal-generation method proposed by LfVoid is not limited to robotics environments and can be used to perform general-purpose editing. We include several general editing results in Figure 4 of the attachment. The results show that LfVoid can perform more **localized** editing as well as **preserve the background** according to only text instructions when compared to existing editing methods. We only choose to study LfVoid’s performance in the robotics context because the ability to perform localized editing with high fidelity to the original image is important when using large-scale text2image models to guide robot learning.


**Prompt Tuning for the editing instructions**

Prompt tuning is very lightweight for LfVoid as we simply visualize the attention maps and see which prompt results in an attention map that better associates each token with its region of interest. We only searched about 5 different prompts for LfVoid, and we also performed the **same** amount and type of prompt tuning for InstructPix2Pix and Imagic. We will include this description in the revised paper.


**Reinforcement Learning with generated visual goal baselines**

At the heart of this work, LfVoid is leveraging knowledge in large-scale diffusion models to generate realistic goals to guide robot learning without any in-domain training. Although the mentioned works [1][2] also use imagined visual goals, these visual goals are generated from a random policy and sampled, and thus cannot be explicitly aligned with human intentions. Furthermore, a human-designed goal image is needed at test time. In comparison, LfVoid allows users to directly describe the goal state through language, and can clearly visualize the goal state to avoid ambiguity.

We thank the reviewers for pointing out this line of work and will add it to the related work section in our final version. We provide an additional comparison with these works and the results can be found in table 5. Since the random exploration policy never reached the desired goal, these works failed to accomplish any of the simulated tasks.


**Additional tasks and results**

Our conducted experiments cover a wide range of manipulation tasks and visual appearances. For example, the Push task and the Wipe task require distinct skills to achieve the goal image. We also include both simulated environments and real robot tasks to demonstrate LfVoid's feasibility.

During the rebuttal, we have added a distractor object to push-sim and introduced three real-world tasks using the UR5 robot arm and reported the goal generation results in Figure 3 in the attachment.


We kindly refer reviewers to the attached PDF for additional figures and tables.

[1] Nair, Ashvin V., et al. “Visual reinforcement learning with imagined goals.” Advances in neural information processing systems 31 (2018).

[2] Nair, Suraj, Silvio Savarese, and Chelsea Finn. “Goal-aware prediction: Learning to model what matters.” International Conference on Machine Learning. PMLR, 2020.

---

### Decision · Program_Chairs · 2023-09-21

**Decision:**

Accept (poster)

**Comment:**

After discussion with the authors, all reviewers recommend acceptance and the AC agrees. Authors are encouraged to include clarifications and details about hyper-parameter / prompt tuning protocols in the evaluation. The comparison to existing RL with visual goals work is useful exposition that should be included as well even though most of these require the goal image to be specified. Likewise, the additional real-world experiments are valuable.